# Atherosclerotic plaque development in mice is enhanced by myeloid ZEB1 downregulation

M. C. Martinez-Campanario [1], Marlies Cortés [1,18], Alazne Moreno-Lanceta [2,18], Lu Han [1], Chiara Ninfali[1], Verónica Domínguez[3], María J. Andrés-Manzano[4,5], Marta Farràs[6,7], Anna Esteve-Codina[8], Carlos Enrich[2,9], Francisco J. Díaz-Crespo[10], Belén Pintado[3], Joan C. Escolà-Gil [6,7], Pablo García de Frutos [5,11,12], Vicente Andrés[4,5], Pedro Melgar-Lesmes[2,13,14,15] & Antonio Postigo [1,14,16,17] ✉

Accumulation of lipid-laden macrophages within the arterial neointima is a critical step in atherosclerotic plaque formation. Here, we show that reduced levels of the cellular plasticity factor ZEB1 in macrophages increase atherosclerotic plaque formation and the chance of cardiovascular events. Compared to control counterparts ($Zeb1^{WT}/Apoe^{KO}$), male mice with $Zeb1$ ablation in their myeloid cells ($Zeb1^{\Delta M}/Apoe^{KO}$) have larger atherosclerotic plaques and higher lipid accumulation in their macrophages due to delayed lipid traffic and deficient cholesterol efflux. $Zeb1^{\Delta M}/Apoe^{KO}$ mice display more pronounced systemic metabolic alterations than $Zeb1^{WT}/Apoe^{KO}$ mice, with higher serum levels of low-density lipoproteins and inflammatory cytokines and larger ectopic fat deposits. Higher lipid accumulation in $Zeb1^{\Delta M}$ macrophages is reverted by the exogenous expression of $Zeb1$ through macrophage-targeted nanoparticles. In vivo administration of these nanoparticles reduces atherosclerotic plaque formation in $Zeb1^{\Delta M}/Apoe^{KO}$ mice. Finally, low ZEB1 expression in human endarterectomies is associated with plaque rupture and cardiovascular events. These results set ZEB1 in macrophages as a potential target in the treatment of atherosclerosis.

Atherosclerosis is an inflammatory and degenerative disease of the arterial wall characterized by lipid accumulation and immune cell infiltration within developing atherosclerotic plaques. Advanced plaques with necrosis, a large lipid core, and a thin fibrotic cap are less stable and prone to rupture, a process that can trigger thrombosis and associated cardiovascular accidents (e.g., myocardial infarction, stroke)[1,2].

The landscape of immune cells and their interplays within the atherosclerotic plaque have begun to be uncovered at the transcriptomic and epigenetic levels (refs. 3–8 and reviewed in refs. 9,10). Macrophages are key players in the initiation, progression and com-plications of atherosclerotic plaques[9,11]. Chemokines released from inflamed endothelium and oxidized lipoproteins within the arterial wall attract circulating monocytes, which transmigrate into the intima, differentiate into macrophages, and proliferate to sustain the plaque's macrophage pool[11,12]. As the most plastic immune cells, macrophages can have both proatherogenic and antiatherogenic roles depending on the stage of plaque progression. Modulation of macrophage plasticity is being investigated as a potential therapeutic approach for atherosclerosis[11–13].

Enhanced uptake of oxidized low-density lipoprotein (oxLDL) and/or diminished cholesterol efflux by macrophages and vascular

smooth muscle cells leads to the intracellular buildup of esterified cholesterol, resulting in the formation of foam cells[11,14,15]. Once oxLDL is internalized, lipoproteins move through the endocytic system from early endosomes to late endosomes and lysosomes where cholesterol esters from oxLDL are hydrolyzed to yield free cholesterol. Cholesterol is subsequently transported out of lysosomes by different mechanisms involving, *inter alia*, NPC1, NPC2, and components of the Golgi-associated retrograde protein (GARP) complex that target NPC2 to lysosomes[16–18]. Free cholesterol can be then exported out of the cell using ATP-binding cassette transporters ABCA1 and ABCG1[19,20]. Intracellular accumulation of esterified cholesterol and the formation of cholesterol crystals in macrophages within the arterial intima activate inflammatory pathways[14,15,18,21]. In addition to localized inflammation of the arterial wall, low-grade systemic inflammation in the context of obesity, dyslipidemia and metabolic syndrome, triggers endothelial dysfunction and promotes vulnerable atherosclerotic plaques[22–24].

The transcription factor ZEB1 is best known for triggering an epithelial-to-mesenchymal transition (EMT) in cancer cells[25–28]. In recent years, ZEB1 has emerged as a key regulator of cell plasticity that plays important functions in non-cancer cells. In macrophages, ZEB1's functions vary depending on the context; for instance, in tumor-associated macrophages, it induces both pro-inflammatory and anti-inflammatory genes; in macrophages infiltrating injured tissues, it expedites the transition of macrophages from a pro-inflammatory phenotype to an anti-inflammatory one; and in cytomegalovirus-infected macrophages, it promotes dedifferentiation, loss of archetypal functions, and the acquisition of stem cell-like traits[29–31].

These findings prompted us to study the potential role of ZEB1 in macrophages in the context of atherosclerosis. The study uses human endarterectomies and a mouse model where *Zeb1* expression is specifically knocked out in myeloid cells and that is later crossed with the atherosclerosis-prone *Apoe*[KO] mouse. We find that *Zeb1* ablation reduces cholesterol efflux and increases intracellular lipid accumulation, leading to the formation of larger atherosclerotic plaques with less fibrosis and more necrosis. Analysis of human samples shows that low ZEB1 expression correlates with less stable plaques and higher incidence of cardiovascular events.

## Results

### Ubiquitous downregulation of *Zeb1* increases plaque formation

The *Stockholm–Tartu Atherosclerosis Reverse Network Engineering Task* (STARNET) study integrated RNA-seq, genetic and clinical data to identify gene-regulatory coexpression networks and "key drivers" within each network in the atherosclerotic aortic root and other tissues from both patients with and without coronary artery disease[32]. Our analysis of the STARNET database (http://starnet.mssm.edu/) revealed that ZEB1 is a key driver in one of the coexpression networks in the study when comparing the aortic root of individuals with coronary artery disease to controls (Supplementary Fig. S1a).

To investigate a potential role for ZEB1 in atherosclerosis, we first examined plaque formation in a constitutive *Zeb1*-deficient mouse. *Zeb1*(−/−) mice are embryonically lethal[33]. However, *Zeb1* downregulation in all cells of the *Zeb1*(+/−) mice to around half of the levels in wild-type counterparts [referred to hereafter as *Zeb1*(+/+) mice] is sufficient to block ZEB1 functions in multiple systems (e.g., tumor progression, muscle wasting, muscle injury and regeneration) and cell types, including macrophages[29,30,34–38]. *Zeb1*(+/+) and *Zeb1*(+/−) mice were crossed with the *Apoe*[KO] mouse[39], a well-established mouse model to study atherosclerosis, to generate *Zeb1*(+/+)/*Apoe*[KO] and *Zeb1*(+/−)/*Apoe*[KO] mice (Supplementary Fig. S1b). In *Zeb1*(+/−)/*Apoe*[KO] mice, macrophages expressed half the *Zeb1* expression compared to *Zeb1*(+/+)/*Apoe*[KO] mice (Supplementary Fig. S1c).

When *Apoe*[KO] mice are subjected to a high cholesterol/fat diet (commonly referred to as "Western diet"), they develop obesity,

hypercholesterolemia, atherosclerosis, glucose resistance, and liver steatosis[39,40]. *Zeb1*(+/+)/*Apoe*[KO] and *Zeb1*(+/−)/*Apoe*[KO] mice were fed *at libitum* with a regular Chow diet for their first 8 weeks and then switched to the Western diet for the following 12 weeks (Fig. 1a). Compared to wild-type counterparts, whole-body downregulation of *Zeb1* in *Zeb1*(+/−)/*Apoe*[KO] mice increased body weight and the size of atherosclerotic plaques in the aortic sinus (Fig. 1b, c). Altogether, these data suggested that ZEB1 has an anti-atherogenic function.

### Downregulation of *Zeb1* expression in myeloid cells increases plaque formation

To test whether the results in Fig. 1b, c were mediated by ZEB1 expression in myeloid cells, we generated *Zeb1*[fl/fl] mice (referred to hereafter as *Zeb1*[WT] mouse) and crossed them with *LysM*[Cre] mice to delete *Zeb1* in myeloid cells (*Zeb1*[ΔM] mouse) (Supplementary Fig. S1d). *Zeb1* expression in *Zeb1*[ΔM] macrophages was reduced by more than 90% compared to that in *Zeb1*[WT] macrophages (Supplementary Fig. S1e). To examine the effect of ZEB1 in atherosclerosis development, *Zeb1*[WT] and *Zeb1*[ΔM] mice were each crossed with *Apoe*[KO] mice to generate the two experimental models used in the rest of the study, namely, *Zeb1*[WT]/*Apoe*[KO] and *Zeb1*[ΔM]/*Apoe*[KO] mice (Supplementary Fig. S1d). Mice were fed *at libitum* with the Chow diet for the first 8 weeks and with the Western diet for the following 10 weeks (Fig. 1d). *Zeb1* expression was also reduced in *Zeb1*[ΔM]/*Apoe*[KO] macrophages compared to that in *Zeb1*[WT]/*Apoe*[KO] macrophages (Fig. 1e and Supplementary Fig. S1f).

*En face* staining of the entire aortas with Oil Red O (ORO), which detects neutral lipids and cholesteryl esters, showed larger atherosclerotic plaques in *Zeb1*[ΔM]/*Apoe*[KO] mice compared to *Zeb1*[WT]/*Apoe*[KO] counterparts (Fig. 1f), a result that was confirmed by staining of aortic root cross-sections with hematoxylin and eosin (H&E) (Fig. 1g). In *Zeb1*[ΔM]/*Apoe*[KO] mice, atherosclerotic plaques had increased lipid accumulation compared to *Zeb1*[WT]/*Apoe*[KO] counterparts, as evaluated by Oil Red O (ORO) staining (Fig. 1h) and confirmed by BODIPY™ 493/503 (hereinafter referred to as BODIPY), which also stains neutral lipids (Fig. 1i). Importantly, macrophage infiltration was greater in *Zeb1*[ΔM]/*Apoe*[KO] plaques than in *Zeb1*[WT]/*Apoe*[KO] plaques (Fig. 1j). *Zeb1*[ΔM]/*Apoe*[KO] plaques also displayed less fibrosis than *Zeb1*[WT]/*Apoe*[KO] plaques, as revealed by lower collagen deposits assessed by Sirius Red and Masson's Trichrome staining (Fig. 1k and Supplementary Fig. S1g). Evaluation of anuclear and afibrotic necrotic areas in the plaques of both genotypes showed that *Zeb1*[ΔM]/*Apoe*[KO] plaques had more necrosis than *Zeb1*[WT]/*Apoe*[KO] plaques (Fig. 1l). *Zeb1*[ΔM]/*Apoe*[KO] plaques also displayed signs of compromised efferocytosis (a lower ratio of "macrophage-associated" apoptotic cells to "free" apoptotic cells), which has been shown to contribute to intra-plaque necrosis[41] (Fig. 1m). Altogether, these results indicate that ZEB1 elimination in myeloid cells led to increased plaque formation, with more necrosis and less fibrosis within the plaques.

### *Zeb1*[ΔM]/*Apoe*[KO] mice exhibit systemic inflammation and lipid accumulation

We next sought to examine if ZEB1 regulates the phenotype of intra-plaque macrophages. Single-cell RNA-seq (scRNAseq) studies of atherosclerotic plaque macrophages in human patients and mouse models of atherosclerosis have identified three main subpopulations: aortic resident, inflammatory, and foam macrophages (refs. 3–6,8, reviewed in ref. 10). Our analysis of published scRNAseq data from human and mouse atherosclerotic plaques[3–6,8] indicated that ZEB1 is expressed at higher levels in inflammatory macrophages than in other macrophage subpopulations (Fig. 2a and Supplementary Fig. S2a).

We used flow cytometry to examine the shares of total macrophages (CD11b[+] F4/80[+]), inflammatory macrophages (CD86[+]), and foam macrophages (CD9[+]), out of total CD45[+] immune cells isolated from the plaques of *Zeb1*[WT]/*Apoe*[KO] and *Zeb1*[ΔM]/*Apoe*[KO] mice. In line with

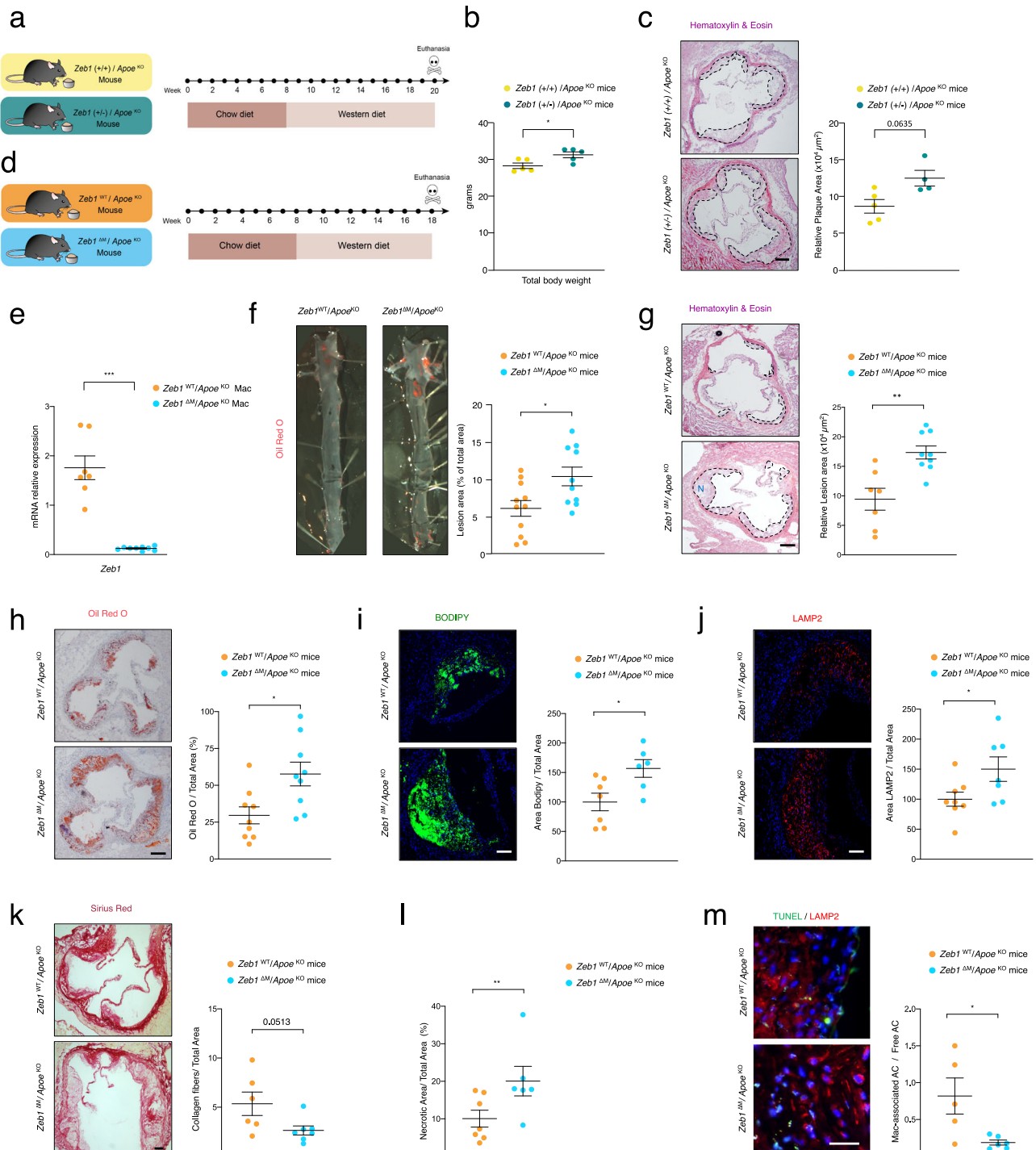

**Fig. 1 | Downregulation of *Zeb1* expression in myeloid cells increases plaque formation. a** Feeding protocol schematic for *Zeb1*(+/+)/*ApoE*^KO and *Zeb1*(+/−)/*ApoE*^KO male mice. From birth until 8 weeks of age, mice were provided *ad libitum* access to the Chow diet, followed by 12 weeks on the Western diet. **b** Total bodyweight of *Zeb1*(+/+)/*ApoE*^KO and *Zeb1*(+/−)/*ApoE*^KO mice at the end of the protocol before euthanasia (*n* = 5). **c** Representative captures and quantification of the atherosclerotic lesion area in aortic root sections of *Zeb1*(+/+)/*ApoE*^KO (*n* = 4) and *Zeb1*(+/−)/*ApoE*^KO (n = 5) mice at the end of the feeding protocol. **d** Feeding protocol schematic for *Zeb1*^WT/*Apoe*^KO and *Zeb1*^ΔM/*Apoe*^KO male mice. From birth to the age of 8-week-old, mice were provided *ad libitum* access to the Chow diet followed by 10 weeks on the Western diet. **e** *Zeb1* mRNA levels in the peritoneal macrophages of *Zeb1*^WT/*Apoe*^KO (*n* = 7) and *Zeb1*^ΔM/*Apoe*^KO (*n* = 9) mice. **f** Representative *en face* ORO staining images and atherosclerosis lesion area quantification in *Zeb1*^WT/*Apoe*^KO (*n* = 11) and *Zeb1*^ΔM/*Apoe*^KO (*n* = 10) mice at the end of the Western diet feeding protocol. **g** As in **f**, representative images and quantification of the atherosclerosis lesion area of aortic

root sections stained with H&E from mice of both genotypes. The dashed line delineates the plaque and "N" indicates a representative area of necrosis. Scale bar: 200 μm. (*n* = 7,9). **h** As in **g**, but sections were stained with ORO to assess lipid accumulation. Scale bar: 200 μm. (*n* = 9,9). **i** As in **g**, but sections were stained with BODIPY® 493/503 (hereinafter referred to as BODIPY). Scale bar: 50 μm. (*n* = 7, 6). **j** As in **g**, but sections were stained for LAMP2/MAC3 (clone M3/84, dilution 1/50). Scale bar: 50 μm (*n* = 8,7). **k** As in **g**, but tissue sections were stained for Sirius Red. Scale bar: 100 μm (*n* = 6, 7). **l** Quantification of plaque necrosis from the staining for Sirius Red. (*n* = 6,7). **m** As in **g**, but sections were stained with TUNEL and LAMP2/MAC3 (clone M3/84, dilution 1/50) along with DAPI for nuclear staining. Quantification of "macrophage-associated" apoptotic cells to "free AC". Scale bar: 25 μm. (*n* = 5,6). All Graphs represent mean values ± SEM with two-tailed unpaired Mann–Whitney test. *p* ≤ 0.001 (***), *p* ≤ 0.01 (**) or *p* ≤ 0.05 (*) levels, or non-significant (ns) for values of *p* > 0.05, and with specified numerical *p* values for 0.05 < *p* < 0.075. The raw data, along with *p* values from the statistical analyses, are included in the Source Data file.

the immunostaining data in Fig. 1j, macrophages represented a higher share of infiltrating CD45[+] immune cells in Zeb1[ΔM]/Apoe[KO] plaques than in Zeb1[WT]/Apoe[KO] plaques (Fig. 2b). The share of CD86[+] macrophages out of total macrophages (CD45[+] CD11b[+] F4/80[+]) was similar between both genotypes (Fig. 2c). However, the share of CD86[+] macrophages out of CD45[+] cells was higher in Zeb1[ΔM]/Apoe[KO] mice, consistent with the higher number of intra-plaque macrophages in these mice (Fig. 2d, e). In turn, the share of CD9[+] macrophages in Zeb1[ΔM]/Apoe[KO] plaques was higher compared to Zeb1[WT]/Apoe[KO] plaques both out of all macrophages and out of CD45[+] cells (Fig. 2f–h).

Mice from both genotypes exhibited similar counts of peripheral blood leukocytes, including monocytes (Supplementary Fig. S2b). Similarly, the share of myeloid populations (Cd11c[+], F4/80[+], Gr-1[+], Ly6C[+]) among CD11b[+] cells in the spleen in both genotypes showed no significant differences (Supplementary Fig. S2c).

Zeb1[ΔM]/Apoe[KO] mice exhibited a higher body weight than Zeb1[WT]/Apoe[KO] mice (Fig. 2i). Obesity is both a major risk factor and a marker for metabolic syndrome and atherosclerosis[2,42]. Ectopic deposits of visceral white adipose tissue (WAT) release inflammatory mediators (e.g., classical cytokines, adipokines), which trigger systemic inflammation and endothelial dysfunction, thus promoting plaque formation and fatty liver. The contribution of WAT to systemic inflammation is mediated by both adipocytes and increased infiltration of immune cells, primarily macrophages[42,43]. Compared to Zeb1[WT]/Apoe[KO] mice, the WAT of Zeb1[ΔM]/Apoe[KO] mice exhibited increased infiltration of macrophages (Fig. 2j), which were knocked out for Zeb1 and expressed higher Il1b mRNA levels in comparison to WAT macrophages from Zeb1[WT]/Apoe[KO] mice (Fig. 2k, l).

To assess the overall inflammatory status of Zeb1[WT]/Apoe[KO] and Zeb1[ΔM]/Apoe[KO] mice, we evaluated a panel of inflammatory markers (cytokines and chemokines) in the serum of mice of both genotypes. For this purpose, we used a quantitative bead-based multiplex array, which was subsequently analyzed by flow cytometry (Fig. 2m). Zeb1[ΔM]/Apoe[KO] mice exhibited higher serum levels of inflammatory cytokines IL1β, IL2, and IL12 cytokines along with the chemotactic chemokine CCL2, in comparison with Zeb1[WT]/Apoe[KO] mice. These results suggest an exacerbated systemic inflammatory status in Zeb1[ΔM]/Apoe[KO] mice. We also found that serum levels of leptin were higher in Zeb1[ΔM]/Apoe[KO] mice than in Zeb1[WT]/Apoe[KO] mice (Fig. 2n). Produced primarily by WAT adipocytes, leptin promotes monocyte recruitment to plaques and macrophage secretion of pro-inflammatory cytokines[44–46].

Disrupted glucose metabolism, as seen in conditions like pre-diabetes and type 2 diabetes, as well as impaired glucose tolerance following glucose loading, are linked to an elevated risk of carotid atherosclerosis[47]. Although Western diet-fed mice from both genotypes had similar levels of serum glucose (Supplementary Fig. S2d), fasted Zeb1[ΔM]/Apoe[KO] mice subjected to a glucose load exhibited higher serum glucose levels and a delayed return to basal levels than Zeb1[WT]/Apoe[KO] mice (Fig. 2o and Supplementary Fig. S2e).

Increased secretion of proinflammatory cytokines and adipokines from the adipose tissue are also key contributors to the development of non-alcoholic fatty liver disease (NAFLD) and non-alcoholic steatohepatitis (NASH)[48,49]. At the end of the 10 weeks protocol on the Western diet, Zeb1[ΔM]/Apoe[KO] mice exhibited heavier livers with increased lipid deposits compared to Zeb1[WT]/Apoe[KO] mice (Fig. 2p–r), suggesting that Zeb1 ablation in macrophages has other systemic effects beyond aggravating atherosclerosis. However, the increased lipid accumulation in the liver of Zeb1[ΔM]/Apoe[KO] is likely to be an indirect effect of the higher systemic inflammation, given that liver macrophages from both genotypes expressed comparable levels of Zeb1 (Supplementary Fig. S2f).

The transcription factor SREBP1c triggers the activation of liver enzymes that promote de novo lipogenesis, leading to the accumulation of fatty acids in the liver and higher lipid serum levels (reviewed in[49,50]). Compared to Zeb1[WT]/Apoe[KO] livers, the livers of Zeb1[ΔM]/Apoe[KO]

mice exhibited higher expression of the nuclear processed form of SREBP1c (Fig. 2s). Compared to Zeb1[WT]/Apoe[KO] mice, Zeb1[ΔM]/Apoe[KO] mice also exhibited higher serum levels of total cholesterol and LDL, and lower levels, although not statistically significant, of HDL (Fig. 2t). There were no differences in the levels of serum triglycerides between both genotypes. (Supplementary Fig. S2g).

Increased deposits of lipids in Zeb1[ΔM]/Apoe[KO] were not limited to the arteries and liver. Staining with ORO in the kidneys and skeletal muscles of mice of both genotypes revealed an increase of lipid deposits in both tissues in Zeb1[ΔM]/Apoe[KO] mice, although the increase was not statistically significant in the kidneys (Supplementary Fig. S2h and S2i).

### ZEB1 inhibits lipid accumulation in macrophages

The potential role of ZEB1 in the regulation of intracellular lipid accumulation in macrophages was investigated using fluorescence staining for BODIPY and Filipin, two fluorescent dyes that stain for all types of neutral lipids and unesterified cholesterol, respectively[51]. Zeb1[WT] and Zeb1[ΔM] macrophages were either untreated or incubated with oxLDL for 24 h. Untreated Zeb1[WT] and Zeb1[ΔM] macrophages displayed similar levels of BODIPY and Filipin staining, but oxLDL induced higher staining for both dyes in Zeb1[ΔM] macrophages (Fig. 3a). Macrophages isolated from Zeb1[ΔM]/Apoe[KO] mice fed with Western diet displayed higher BODIPY staining compared to their Zeb1[WT]/Apoe[KO] counterparts (Fig. 3b). These results indicate that Zeb1[ΔM] macrophages accumulate more lipids than Zeb1[WT] counterparts, consistent with the in vivo atherogenic effect of eliminating Zeb1 in macrophages.

### ZEB1 promotes cholesterol efflux in macrophages through upregulation of ABCA1 and ABCG1

The increased lipid accumulation in Zeb1[ΔM] macrophages prompted us to investigate whether ZEB1 modulates lipid influx and/or efflux. To assess this, we first examined the uptake of oxLDL labeled with the fluorescent dye Atto-655 using flow cytometry. Zeb1[ΔM] macrophages displayed similar or only slightly higher (albeit non-significant) Atto-665-oxLDL uptake than Zeb1[WT] counterparts (Supplementary Fig. S3a). Although Zeb1[ΔM] macrophages expressed higher mRNA levels of the cholesterol influx scavenger receptor Cd36 (Scarb3) than Zeb1[WT] macrophages in the absence of oxLDL, oxLDL upregulated Cd36 expression in macrophages of both genotypes to similar levels (Supplementary Fig. S3b). The scavenger receptor SR-B1 (encoded by Scarb1), which participates in both cholesterol uptake and efflux[52], was expressed at comparable levels in oxLDL-treated Zeb1[WT] and Zeb1[ΔM] macrophages (Fig. 3c).

Cholesterol can be exported out of cells via the ATP-binding cassette transporters ABCA1 and ABCG1. Efflux to lipid-free apolipoprotein A1 (APOA1) requires ABCA1, whereas efflux to HDL requires ABCG1[19,20]. Under basal conditions, macrophages of both genotypes exhibited comparable expression levels of Abca1 and Abcg1 (Fig. 3c). However, while oxLDL led to an upregulation of both efflux transporters in Zeb1[WT] macrophages, it had no effect on Zeb1[ΔM] macrophages (Fig. 3c). To quantify the cholesterol efflux capacity of oxLDL-treated Zeb1[WT] and Zeb1[ΔM] macrophages, [3]H-labeled cholesterol was employed. In Zeb1[ΔM] macrophages, the efflux of [3]H-cholesterol to both lipid-free APOA1 and HDL was reduced to approximately half that observed in Zeb1[WT] macrophages (Fig. 3d). Collectively, these data suggest that increased lipid accumulation in Zeb1[ΔM] macrophages is, at least in part, attributable to their diminished cholesterol efflux capacity.

### ZEB1 inhibits lipid accumulation in macrophages through activation of AMPK signaling

Next, we investigated whether ZEB1 modulates the expression of pathways and genes that regulate lipid metabolism. AMPK/LXRα

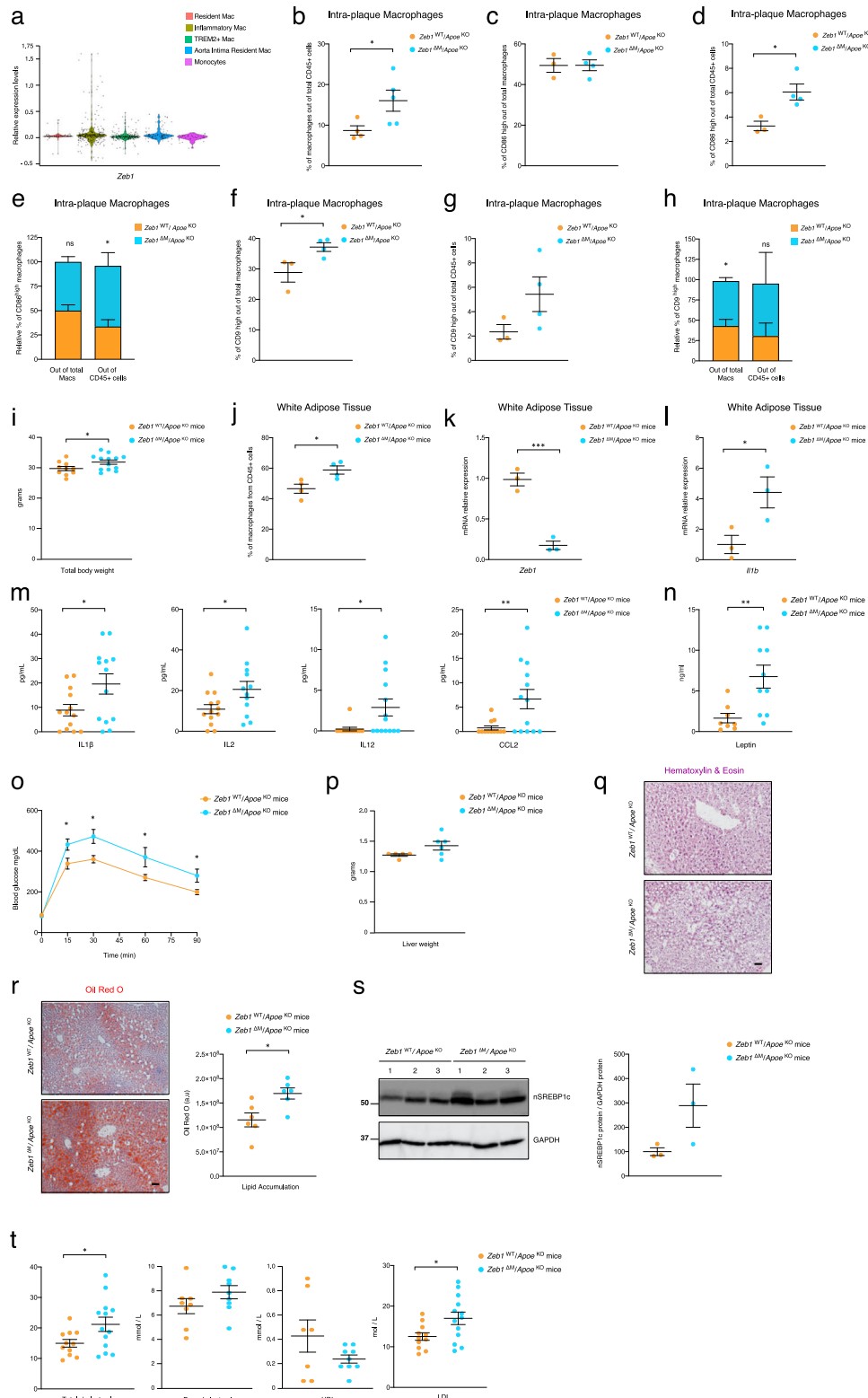

signaling plays a protective role in atherosclerosis[53]. LXRα (encoded by *Nr1h3*), activates cholesterol efflux by upregulating the expression of ABCA1 and ABCG1[53,54].

Relative to oxLDL-*Zeb1*[WT] counterparts, oxLDL-*Zeb1*[ΔM] macrophages exhibited lower levels of *Prkaa1*, the catalytic subunit of AMPK, which functions as a cellular energy sensor[55] (Fig. 3e). *Prkaa1* expression was further downregulated upon oxLDL treatment. Macrophages from both genotypes expressed similar levels of *Nr1h3* in the absence of oxLDL; however, oxLDL led to a reduction in *Nr1h3* expression in

*Zeb1*[ΔM] macrophages, while it had no impact on *Zeb1*[WT] macrophages (Fig. 3e). The expression of *Ppargc1a* (which regulatest mitochondrial biogenesis and oxidative phosphorylation) was found to be lower in *Zeb1*[ΔM] macrophages, particularly in oxLDL-treated cells (Fig. 3e). The expression of glucose transporter *Slc2a1* in *Zeb1*[ΔM] macrophages was higher in oxLDL-treated *Zeb1*[ΔM] macrophages than in oxLDL-treated *Zeb1*[WT] macrophages (Fig. 3e).

The diminished expression of AMPK targets (*Nr1h3, Ppargc1a*) in oxLDL-treated *Zeb1*[ΔM] macrophages in comparison to *Zeb1*[WT]

**Fig. 2 | *Zeb1*[ΔM]/*Apoe*[KO] mice exhibit systemic inflammation and fat accumulation.** **a** Macrophage subpopulations were analyzed using published scRNAseq datasets of mouse atherosclerosis (GSE116240 and GSE149070)[3,4,11]. **b** The share of macrophages out of CD45[+] cells within the atherosclerotic plaque of *Zeb1*[WT]/*Apoe*[KO] (*n* = 4) and *Zeb1*[ΔM]/*Apoe*[KO] (*n* = 5) mice was determined by FACS. **c** As in **b**, but the share of CD86[high] macrophages was calculated out of the total number of macrophages (CD45[+], CD11b[+], F4/80[+]). (*n* = 3,4). **d** As in **b**, but the share of CD86[high] macrophages was calculated out of total CD45[+] cells. (*n* = 3, 4). **e** Stacked bar chart for (**c**) and (**d**). **f–h** As in **c–e**, but for CD9[high] macrophages. **i** Total body weight of *Zeb1*[WT]/*Apoe*[KO] (*n* = 10) and *Zeb1*[ΔM]/*Apoe*[KO] (*n* = 13) mice at the end of the feeding protocol. **j** The share of macrophages out of CD45[+] cells infiltrating the WAT of *Zeb1*[WT]/*Apoe*[KO] and *Zeb1*[ΔM]/*Apoe*[KO] mice. (*n* = 4). **k** and **l** *Zeb1* and *Il1b* mRNA levels in the macrophages infiltrating the WAT of *Zeb1*[WT]/*Apoe*[KO] and *Zeb1*[ΔM]/*Apoe*[KO] mice (*n* = 3). **m** Serum levels of IL1β (*n* = 13), IL2 (*n* = 13,12), IL12 (*n* = 12,14), and CCL2 (*n* = 12,13) from *Zeb1*[WT]/*Apoe*[KO] and *Zeb1*[ΔM]/*Apoe*[KO] mice at the end of the feeding protocol. **n** Serum levels of leptin in *Zeb1*[WT]/*Apoe*[KO] (*n* = 8) and *Zeb1*[ΔM]/*Apoe*[KO] (*n* = 10) mice. **o** Glucose tolerance test in *Zeb1*[WT]/*Apoe*[KO] (*n* = 7) and *Zeb1*[ΔM]/*Apoe*[KO] (*n* = 6) mice at the end of the feeding protocol plus 16 h of fasting. **p** Liver weight of *Zeb1*[WT]/*Apoe*[KO] (*n* = 5) and *Zeb1*[ΔM]/*Apoe*[KO] (*n* = 6) mice. **q** Representative images of liver sections from *Zeb1*[WT]/*Apoe*[KO] and *Zeb1*[ΔM]/*Apoe*[KO] mice stained with H&E. Scale bar: 20 μm. **r** As in **q**, sections were stained for ORO. Representative images and quantification. Scale bar: 20 μm. (*n* = 6). **s** Lysates from the livers of *Zeb1*[WT]/*Apoe*[KO] and *Zeb1*[ΔM]/*Apoe*[KO] at the end of the feeding protocol plus 12 h of fasting were blotted for mSREBP1c (clone 2A4, dilution 1/1000) and GAPDH (1E6D9, 1/20,000) as loading control. (*n* = 3). **t** Serum levels of total cholesterol (*n* = 11,13), free colesterol (*n* = 8,9), HDL (*n* = 7,9), and LDL (*n* = 11,13) in *Zeb1*[WT]/*Apoe*[KO] and *Zeb1*[ΔM]/*Apoe*[KO] mice at the end of the feeding protocol. Graphs represent mean values ± SEM with two-tailed unpaired Student's *t* test. *p* ≤ 0.001 (***), *p* ≤ 0.01 (**) or *p* ≤ 0.05 (*) levels, or non-significant (ns) for values of *p* > 0.05. The raw data, along with *p* values from statistical analyses are included in the Source Data file.

macrophages led us to investigate whether the elevated lipid accumulation in the former was associated with impaired activation of the AMPK signaling pathway. *Zeb1*[WT] and *Zeb1*[ΔM] macrophages were untreated or treated for 24 h with oxLDL, either in the presence or absence of the AMPK activator A769662. BODIPY staining showed that AMPK activation reduced lipid accumulation in oxLDL-treated *Zeb1*[ΔM] macrophages to similar levels as in *Zeb1*[WT] macrophages (Fig. 3f). Treatment with A769662 also increased the expression of *Nr1h3* and *Ppargc1a* in *Zeb1*[ΔM] macrophages to levels comparable to those observed in *Zeb1*[WT] macrophages (Fig. 3g). Conversely, inhibition of AMPK signaling with the pyrazolopyrimidine Compound C, increased lipid content in oxLDL-treated *Zeb1*[WT] macrophages but had no effect in oxLDL-treated *Zeb1*[ΔM] macrophages (Fig. 3h). These results suggest that ZEB1 expression inhibits lipid accumulation, at least in part, through activation of AMPK signaling.

## ZEB1 regulates the expression of genes involved in the intracellular trafficking of lipids

We next sought to further define the gene signature by which ZEB1 regulates macrophage lipid accumulation and atherosclerotic plaque formation. To this end, we isolated macrophages from the aortic sinus plaques of *Zeb1*[WT]/*Apoe*[KO] and *Zeb1*[ΔM]/*Apoe*[KO] mice after the 10-weeks period with Western diet and used RNA sequencing (RNAseq) to study their transcriptome. Analysis of the most differentially expressed genes (DEGs) between macrophages of both genotypes revealed that *Zeb1*[ΔM]/*Apoe*[KO] macrophages exhibited increased expression of *Aldh3b1* (associated with retinoic acid metabolism), *Rnase4* (an M2-like macrophage marker), and *Trappc3* (involved in intracellular trafficking) (Fig. 4a). In turn, DEGs downregulated in *Zeb1*[ΔM]/*Apoe*[KO] macrophages included *Gab3* (macrophage differentiation), *Vsp52* (intracellular vesicle trafficking), *Adss* (purine metabolism), *Asl* (involved in nitric oxide production) and *Chpm1b* (lipid droplets, fatty acids trafficking and lysosome membrane repair) (Fig. 4a).

The differential expression of *Vps52*, *Asl*, *Chmp1b*, and *Adss1* in the RNAseq was confirmed by qRT-PCR in macrophages of both genotypes in the presence or absence of oxLDL (Fig. 4b). Notably, several of the dysregulated DEGs in *Zeb1*[ΔM]/*Apoe*[KO] macrophages are involved in organelle and endocytic homeostasis, as well as the intracellular traffic of cholesterol (Supplementary Fig. S4a). For instance, VPS52, a component of the GARP complex, mediates the transfer of NPC2 between the trans-Golgi and lysosomes, being therefore required for cholesterol trafficking[17]. Interestingly, some GARP proteins also appear in the same gene regulatory network as ZEB1 in the STARNET study[32].

The altered expression of genes involved in intracellular traffic prompted us to examine the ultrastructural alterations of the atherosclerotic lesions of Western diet-fed *Zeb1*[WT]/*Apoe*[KO] and *Zeb1*[ΔM]/*Apoe*[KO] mice using transmission electron microscopy (TEM). Macrophages within the atherosclerotic plaques of *Zeb1*[ΔM]/*Apoe*[KO] mice displayed more lipid droplets and cholesterol crystals than in *Zeb1*[WT]/*Apoe*[KO] lesions (Fig. 4c and Supplementary Fig. S4b). Importantly, the accumulation of cholesterol crystals in macrophages causes lysosomal damage and triggers the NLRP3 inflammasome[21,56]. Smooth muscle cells and fibroblasts in the plaques of *Zeb1*[ΔM]/*Apoe*[KO] mice also contained more lipid droplets than those in *Zeb1*[WT]/*Apoe*[KO] plaques. Finally, compared to *Zeb1*[WT]/*Apoe*[KO] counterparts, *Zeb1*[ΔM]/*Apoe*[KO] macrophages showed larger endosomes and lysosomes, which were loaded with lipids and appeared to be undergoing fusion and unable to degrade, potentially contributing to their enhanced formation of cholesterol crystals (Fig. 4c and Supplementary Fig. S4b).

## *Zeb1* deletion hinders the intracellular traffic of lipids

To investigate the potential role of ZEB1 in intracellular cholesterol transport, *Zeb1*[WT] and *Zeb1*[ΔM] macrophages were incubated with LDL conjugated to pHrodo™ Green (pHrodo™ Green-LDL) and monitored its fluorescence over time using confocal microscopy. pHrodo™ Green is weakly or non-fluorescent at neutral pH outside of cells, but its intensity increases as pH declines during the progression and maturation of vesicles from endosomes to lysosomes. During the initial 5-min period, pHrodo™ Green-LDL fluorescence demonstrated comparable levels in both *Zeb1*[WT] and *Zeb1*[ΔM] macrophages (captures in Fig. 4d, e, and Supplementary Movies 1 and 2). However, after 10 min, the fluorescence greatly increased in *Zeb1*[ΔM] macrophages while it remained relatively stable in *Zeb1*[WT] macrophages. This suggests that *Zeb1*[ΔM] macrophages had not only compromised their cholesterol efflux (Fig. 3d), but also that their intracellular trafficking of LDL is hindered.

To further investigate this, macrophages of both genotypes were incubated with oxLDL complexed with the constitutive fluorescent DiI [DiIC18[3]] dye. As expected, the fluorescence of DiI-oxLDL declined as it was transferred to late endosomes/lysosomes to generate free cholesterol. However, the fluorescence declined faster in *Zeb1*[WT] macrophages than *Zeb1*[ΔM] macrophages (Fig. 4f, g and Supplementary Fig. S4c), suggesting again that oxLDL trafficking is restrained in *Zeb1*[ΔM] macrophages.

Downregulation of proteins in the GARP complex inhibits the transfer of NPC2 to lysosomes[17]. We stained oxLDL-treated macrophages with NPC2 along with LAMP1, a lysosomal marker that binds cholesterol as well as NPC1/NPC2[57]. Whereas NPC2 and LAMP1 colocalized in oxLDL-treated *Zeb1*[WT] macrophages, this colocalization was greatly reduced in *Zeb1*[ΔM] macrophages (Fig. 4h and Supplementary Fig. S4d).

To gain further insights into the endocytic pathway in *Zeb1*[ΔM] macrophages, we examined macrophages from both genotypes by TEM after internalizing oxLDL (Fig. 4i). *Zeb1*[WT] macrophages exhibited prototypical multivesicular bodies (MVB, late endosomes), indicative of normal uptake and transport of oxLDL. Instead, consistent with the data in Fig. 4d–g, *Zeb1*[ΔM] macrophages aberrantly accumulate cholesterol in the late endocytic compartment (lysosomes), some in the form

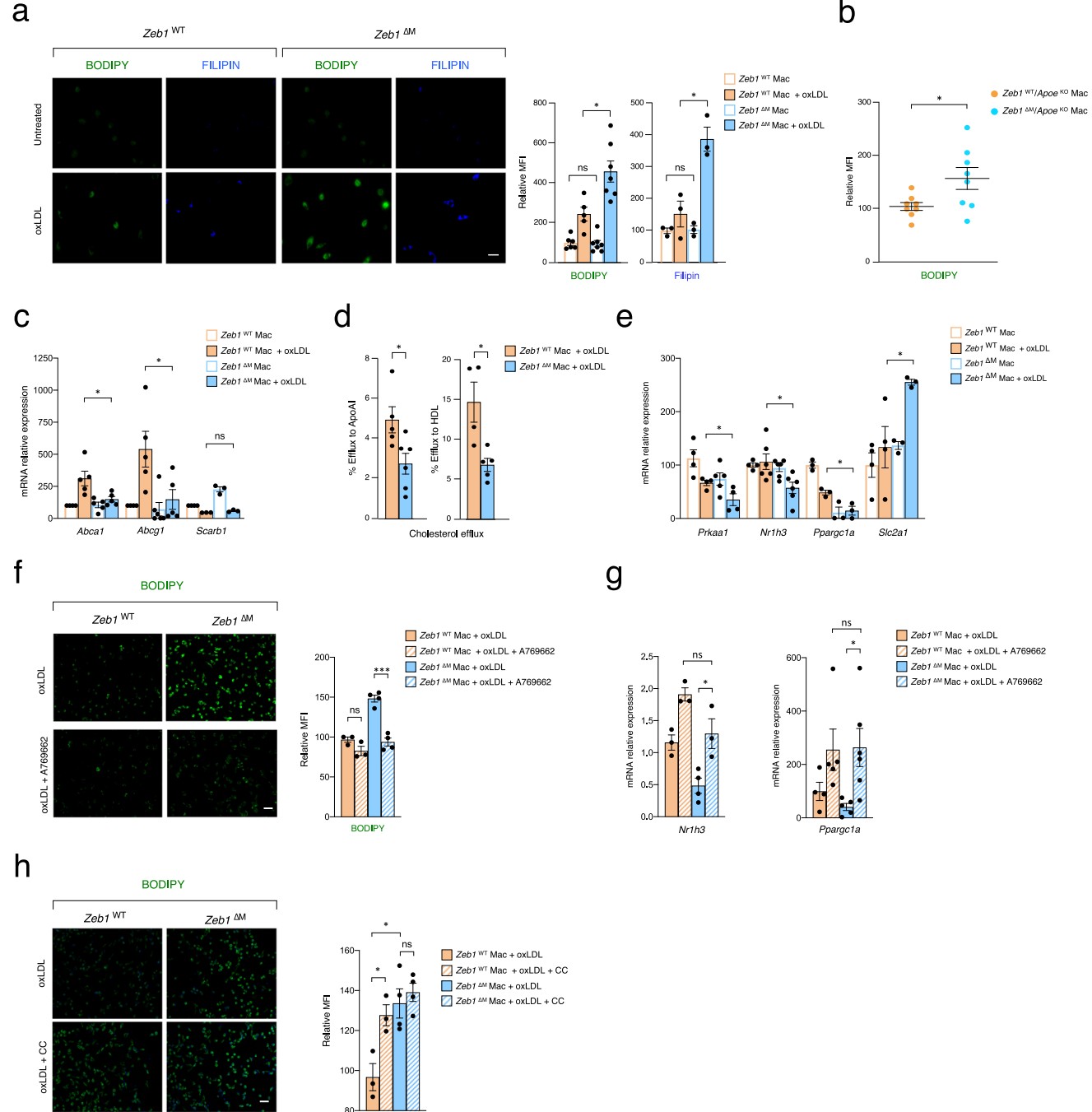

**Fig. 3 | ZEB1 inhibits lipid accumulation and promotes cholesterol efflux in macrophages. a** *Zeb1*WT and *Zeb1*ΔM peritoneal macrophages either untreated or treated with 50 μg/mL of ox-LDL for 24 h were assessed for lipid accumulation with BODIPY and Filipin. Representative images (scale bar: 20 μm) and quantification (BODIPY, *n* = 6,5,7,7; Filipin, *n* = 3). **b** BODIPY staining of macrophages from *Zeb1*WT/*Apoe*KO and *Zeb1*ΔM/*Apoe*KO mice at the end of the feeding protocol (*n* = 8). **c** mRNA levels of *Abca1* (*n* = 4,5,5,5), *Abcg1* (*n* = 4,4,6,5), and *Scarb1* (*n* = 4,3,3,3) were determined by qRT-PCR in *Zeb1*WT and *Zeb1*ΔM peritoneal macrophages untreated or treated for 24 h with 50 μg/mL of oxLDL. **d** Cholesterol efflux to lipid-free APOA1 and HDL in *Zeb1*WT and *Zeb1*ΔM peritoneal macrophages preloaded with ³H-cholesterol and 50 μg/mL of oxLDL (APOA1 *n* = 5,6; HDL *n* = 4,5). **e** As in **c**, but for

*Prkaa1* (*n* = 4,4,5,4), *Nr1h3* (*n* = 4,6,6,6), *Ppargc1a* (*n* = 3), and *Slc2a1* (*n* = 4,4,3,3). **f** BODIPY staining in *Zeb1*WT and *Zeb1*ΔM peritoneal macrophages treated with 50 μg/mL of oxLDL in the presence or absence of 100 μM of A769662. Representative pictures (scale bar: 20 μm) and quantification. (*n* = 3,3,4,4). **g** As in **f**, but *Nr1h3* (*n* = 3,3,4,3) and *Ppargc1a* (*n* = 4,5,5,6) mRNA expression was assessed. **h** As in **f**, but *Zeb1*WT and *Zeb1*ΔM peritoneal macrophages treated with 50 μg/mL of oxLDL in the presence or absence of 15 μM of compound C (CC) (*n* = 3,3,4,4). Graphs represent mean values ± SEM with two-tailed unpaired Student's *t* test. *p* ≤ 0.001 (\*\*\*), *p* ≤ 0.01 (\*\*) or *p* ≤ 0.05 (\*) levels, or non-significant (ns) for values of *p* > 0.05. The raw data, along with *p* values from statistical analyses are included in the Source Data file.

of cholesterol crystals (Fig. 4i, red arrows). *Zeb1*ΔM macrophages also showed signs of Golgi fragmentation (Fig. 4i), which supports an intracellular trafficking malfunction, both in the exocytic pathway and the biogenesis of lysosomes.

Taken together, the data suggest that higher lipid accumulation in *Zeb1*ΔM/*Apoe*KO macrophages and oxLDL-treated *Zeb1*ΔM macrophages result from alterations in the transport of cholesterol through the endocytic system.

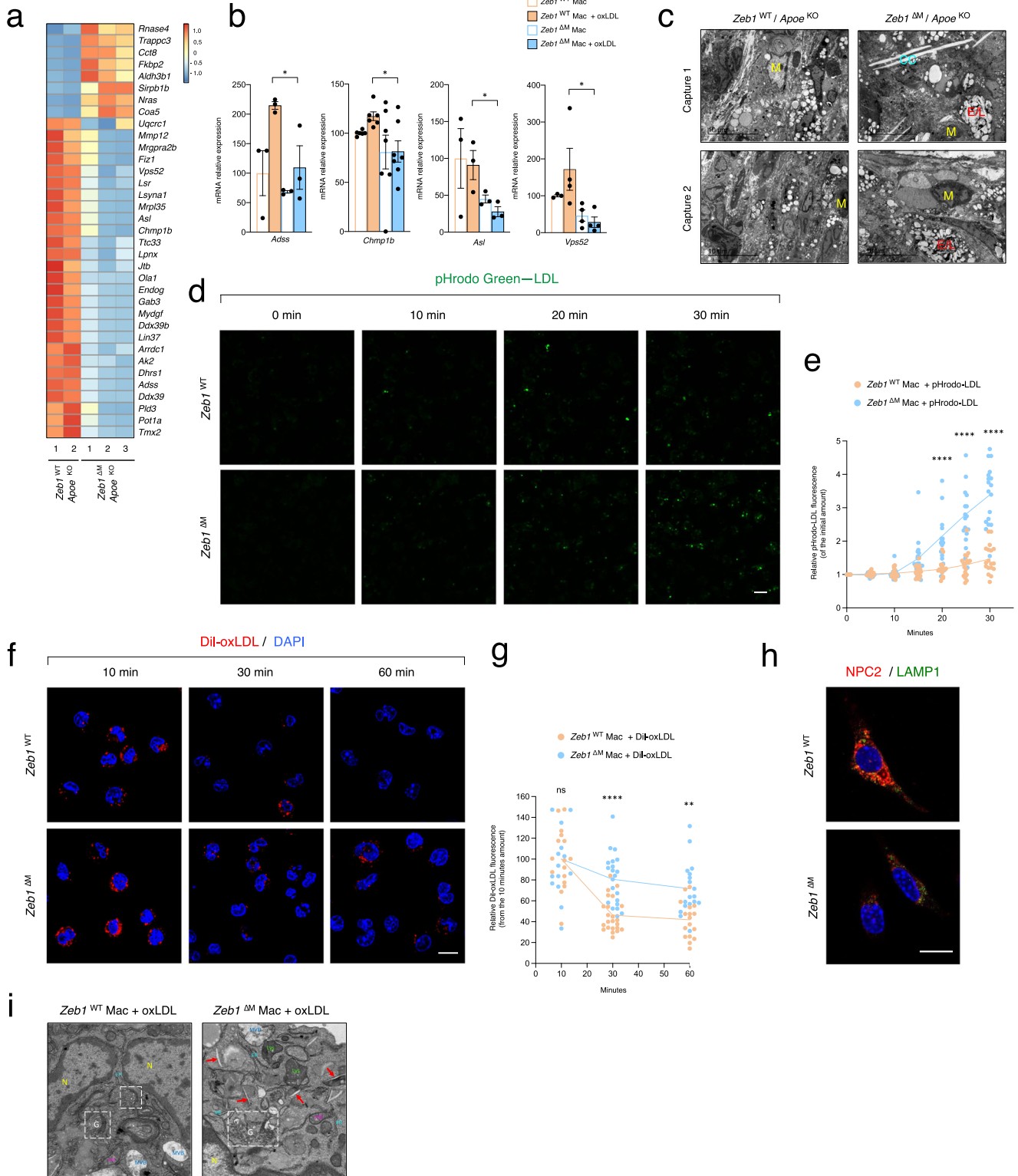

## Macrophage-targeted nanoparticles expressing ZEB1 reduce lipid accumulation and enhance cholesterol efflux

Periodic intravenous administration of nanoparticles to deliver drugs, siRNAs, or cDNAs into intra-plaque macrophages has proven effective in reducing atherosclerosis in mouse models with reduced off-target effects, spurring several clinical trials (refs. 58,59 and reviewed in refs. 60,61). Ongoing advances in bioengineering are continuously improving the biosafety, biocompatibility, and reducing the toxicity and immunogenicity of nanomaterials. Graphene nanostars,

specifically single-wall carbon nanohorns, exhibit higher biocompatibility with macrophages and other cell types than other nanoparticles and showed no signs of toxicity after more than 6 months[62–65]. We explored whether exogenous expression of *Zeb1* using nanoparticles could reverse the deficient cholesterol efflux and enhanced lipid accumulation in *Zeb1*[ΔM] macrophages.

First, we developed dedrimer-graphene nanostars (DGNS)— formed by graphene nanostars (GNS) fused to PAMAM-G5 dendrimers —which have been previously employed for gene delivery in other

**Fig. 4 | Zeb1 deletion hinders the intracellular lipid traffic in macrophages.**
**a** Heatmap of the top differentially expressed genes (DEG) in the RNAseq of infiltrating macrophages in the plaque of Zeb1^WT/Apoe^KO (n = 2) and Zeb1^ΔM/Apoe^KO (n = 3) mice at the end of the Western diet feeding protocol. **b** mRNA levels of Adss1 (n = 3), Chmp1b (n = 6,6,7,7), Asl (n = 3), and VpsS2 (n = 3,4,4,4) were determined by qRT-PCR in Zeb1^WT and Zeb1^ΔM peritoneal macrophages untreated or treated for 24 h with 50 μg/mL of ox-LDL. **c** Ultrastructure of areas containing macrophages in atherosclerotic plaque in Zeb1^WT/Apoe^KO and Zeb1^ΔM/Apoe^KO mice at the end of the feeding protocol were examined by TEM. Two representative captures are shown and additional captures are displayed in Supplementary Fig. S4b. Macrophages (M), cholesterol crystals (CC), late endosomes/lysosomes (E/L). Scale bar: 10 μm (n = 3). **d** The uptake and transport of pHrodo™ Green-LDL fluorescence in Zeb1^WT and Zeb1^ΔM macrophages were captured over time (0–30 min) by confocal microscopy (see the corresponding Supplementary Movies 1 and 2, respectively, in Supplementary Information). Representative captures. Scale bar: 10 μm. **e** As in **d**, n = 18 cells quantification of three independent experiments. **f** DiI-oxLDL fluorescence of Zeb1^WT and Zeb1^ΔM macrophages at 0, 30, and 60 min was assessed by confocal microscopy. See Supplementary Fig. S3c for individual staining pictures.

Representative captures. Scale bar: 10 μm. **g** As in **f**, n = 17,18 cells quantification of two independent experiments. **h** Representative pictures of Zeb1^WT and Zeb1^ΔM macrophages stained for NPC2 (red, clone 19888-1-AP, dilution 1/100) and LAMP1 (green, H4A3, 1/500) and assessed by confocal microscopy of two independent experiments. See Supplementary Fig. S4d for individual staining. Representative captures. Scale bar: 10 μm. **i** Representative TEM images of Zeb1^WT and Zeb1^ΔM macrophages from a 5–6 mice pool for each genotype. In Zeb1^WT macrophages, oxLDL is internalized via receptor-mediated endocytosis, forming multivesicular bodies (MVBs, late endosomes) that fuse with lysosomes to complete degradation. In Zeb1^ΔM macrophages, cholesterol accumulates in endolysosomes and forms crystals (red arrows). Zeb1^ΔM/Apoe^KO macrophages also exhibited Golgi fragmentation (dashed squares). mit mitochondria, MVB multivesicular body (late endosome), G Golgi complex, N nucleus, ER endoplasmic reticulum, Lys lysosome. Scale bar: 500 nm. Graphs represent mean values ± SEM with two-tailed unpaired Student's t test (**b**) or two-way ANOVA test (**e**, **g**). $p \leq 0.001$ (***), $p \leq 0.01$ (**) or $p \leq 0.05$ (*) levels, or non-significant (ns) for values of $p > 0.05$. The raw data, along with p values from statistical analyses are included in the Source Data file.

studies[65]. We left some of the DGNS unlabeled and linked others to FITC (DGNS-FITC) to track their capture by macrophages and their reach to the aortic sinus. Incubation of mouse peritoneal lavage cells with DGNS and DGNS-FITC indicated that virtually all macrophages phagocytosed DGNS-FITC, while almost none of the non-macrophage cells did (Fig. 5a). We also found that intravenous injection of DGNS-FITC efficiently accumulated in the aorta (Fig. 5b).

We next bound DGNS to expression vectors carrying either the Zeb1 cDNA (referred to hereafter as DGNS-Zeb1) or a scrambled non-coding DNA sequence (DGNS-Ctrl). Both expression vectors were driven by the Lyz2 (LysM) promoter, which is the same promoter driving the Cre recombinase in our conditional mouse models. This ensured the specific targeting and overexpression of Zeb1 (or the scrambled DNA) in myeloid cells (Fig. 5c). The physicochemical properties of GNS, DGNS, and the two expression vector-DGNS (DGNS-Ctrl and DGNS-Zeb1) are outlined in the Supplementary Information, and their key parameters are summarized in Supplementary Fig. S5a.

To assess the capacity of DGNS-Zeb1 to exogenously induce ZEB1 expression in macrophages, Zeb1^WT and Zeb1^ΔM macrophages were incubated in vitro with either DGNS-Ctrl or DGNS-Zeb1, in the presence or absence of oxLDL. DGNS-Zeb1, but not DGNS-Ctrl, upregulated Zeb1 in macrophages of both genotypes (Supplementary Fig. S5b).

We investigated whether DGNS-Zeb1 could reverse the impaired functions of Zeb1^ΔM macrophages. Given that ZEB1 is expected to be bound to all available sites on the DNA regulatory regions of its target genes in Zeb1^WT macrophages, the overexpression of Zeb1 in these cells would not have any additional functional effect (see Discussion). Indeed, DGNS-Zeb1 did not increase cholesterol efflux in Zeb1^WT macrophages (Fig. 5d). In contrast, DGNS-Zeb1, but not with DGNS-Ctrl, efficiently rescued the deficient cholesterol efflux of Zeb1^ΔM macrophages to similar levels than in Zeb1^WT macrophages (Fig. 5d). Similarly, DGNS-Zeb1, but not DGNS-Ctrl, effectively reversed the increased lipid accumulation in Zeb1^ΔM macrophages, bringing it to levels comparable to Zeb1^WT macrophages (Fig. 5e). As anticipated, DGNS-Zeb1 nanoparticles had no impact on the intracellular lipid accumulation in oxLDL-treated Zeb1^WT macrophages.

### In vivo use of macrophage-targeted nanoparticles expressing ZEB1 reduces atherosclerotic plaque formation

The in vitro results with DGNS-Zeb1 led us to investigate whether in vivo administration of DGNS-Zeb1 could reduce atherosclerotic plaque formation in Zeb1^ΔM/Apoe^KO mice. Mice were injected intravenously twice weekly with either DGNS-Ctrl or DGNS-Zeb1 throughout the 10-week protocol with Western diet (Fig. 6a). First, we conducted tests to evaluate the in vivo safety and potential

toxicity of DGNS. In comparison to injection with PBS, the administration of DGNS-Ctrl did not lead to any changes in key hematological (total blood count, proportion of individual blood cell populations) and biochemical (serum protein levels, liver transaminases) parameters, and there were no histological alterations observed in liver, kidney, heart, spleen, and lung tissues (Fig. 6b, and Supplementary Fig. S6a–c).

Compared to the administration of DGNS-Ctrl, DGNS-Zeb1 induced ZEB1 expression in macrophages (Fig. 6c). We then examined atherosclerotic plaque formation and hepatic lipid accumulation in Zeb1^ΔM/Apoe^KO mice fed a Western diet and treated with DGNS-Ctrl or DGNS-Zeb1. At the end of the protocol, DGNS-Zeb1 significantly reduced Zeb1^ΔM/Apoe^KO plaque size relative to DGNS-Ctrl, as assessed by ORO en face staining of the entire aorta (Fig. 6d) and H&E and ORO staining of tissue cross-sections of the aortic sinus (Fig. 6e, f). Administration of DGNS-Zeb1, also reduced lipid accumulation in the liver compared to administration of DGNS-Ctrl (Fig. 6g).

### Low levels of ZEB1 in human endarterectomies are associated with a higher incidence of plaque rupture and cardiovascular events

The above results in mouse models led us to hypothesize that low levels of ZEB1 in intra-plaque macrophages of atherosclerotic patients can lead to worse disease outcomes[66,67].

We analyzed ZEB1 expression in a published RNAseq dataset of macrophage-enriched areas from stable and ruptured plaques from human endarterectomies (GSE41571[45]). Stratifying the samples based on whether their ZEB1 expression was above or below the median, we observed that ZEB1 expression was correlated with plaque stability, with all plaques in the ZEB1^high cohort being stable and 83.3% of those in the ZEB1^low cohort being ruptured (Fig. 6h). Similarly, analysis in this array for different markers in samples with plaque stability or rupture, showed that low levels of ZEB1 are associated with higher expression of markers of plaque instability (e.g. inflammatory markers as CCL2, CD38 or LEP) (Fig. 6i).

Neovascularization and intraplaque hemorrhage (IPH) increase the risk of plaque rupture and cardiovascular accidents[66,67]. Our analysis of a publicly available RNAseq dataset of patients with or without intraplaque hemorrhage (IPH) (GSE163154[68]) and the categorization of samples based on their ZEB1 expression above or below the median revealed a strong association between low ZEB1 expression and the presence of IPH (Fig. 6j). Concurrently, in our series of human endarterectomies, the ZEB1^high cohort was associated with asymptomatic patients, whereas the ZEB1^low cohort was associated with individuals who had suffered cardiovascular events (Fig. 6k).

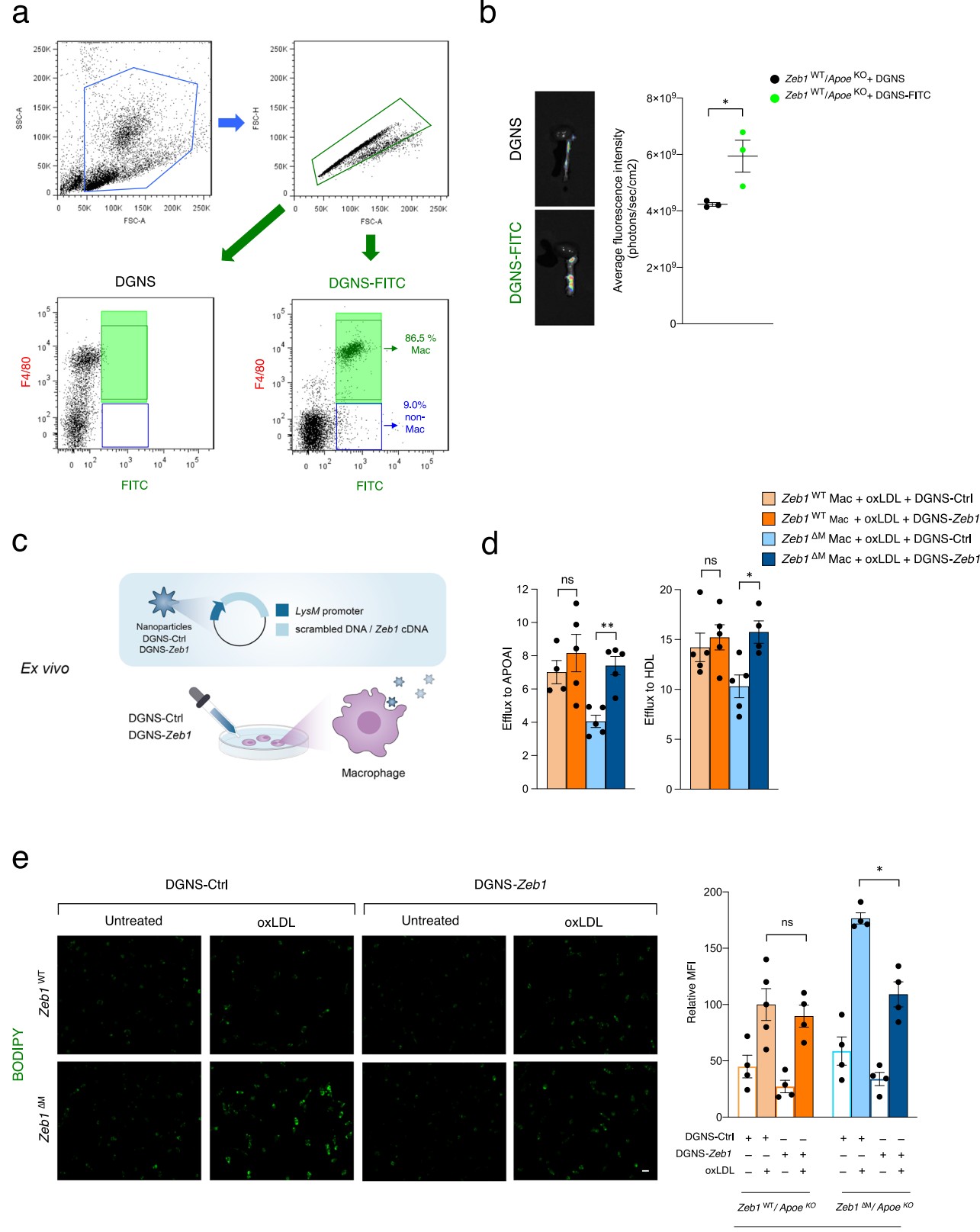

Altogether, our findings indicate that ZEB1 expression in macrophages reduces the size and is associated with increased stability of the atherosclerotic plaque in preclinical models, and suggest that its overexpression in macrophages of patients with low levels of ZEB1 can have a potential therapeutic effect.

## Discussion

Macrophages play a key role in atherosclerosis[11]. Here we reported that ZEB1 elimination in macrophages augments plaque size through, at least in part, reduced cholesterol efflux. ZEB1 has been extensively characterized for its role promoting a dedifferentiated and stem-like

**Fig. 5 | Macrophage-targeted nanoparticles expressing ZEB1 reduce lipid accumulation and enhance cholesterol efflux. a** Peritoneal cells were incubated for 6 h with DGNS and DGNS-FITC and the identity of macrophages versus non-macrophages was assessed by their expression of F4/80. **b** Intravenous injection of DGNS or DGNS-FITC into the tail of *Apoe*[KO] mice and analysis of FITC fluorescence. Representative pictures and quantification of FITC fluorescence using ImageJ software (*n* = 3). **c** Schematic of the generation of DGNS-Ctrl and DGNS-*Zeb1* and ex vivo administration into macrophages. **d** Macrophages from both genotypes were assessed for cholesterol efflux as in Fig. 3d. (APOAI *n* = 4,5,5,5; HDL *n* = 5,5,5,4). **e** As in **d**, but macrophages from both genotypes were assessed for BODIPY staining (*n* = 4). Representative captures (scale bar: 20 μm) and quantification of four independent experiments. Graphs represent mean values ± SEM with two-tailed unpaired Student's t test (**b**) or two-tailed unpaired Mann-Whitney (**d**, **e**). *p* ≤ 0.001 (***), *p* ≤ 0.01 (**) or *p* ≤ 0.05 (*) levels, or non-significant (ns) for values of *p* > 0.05. Raw data along with *p* values from for statistical analyses are included in the Source Data file.

phenotype in cancer cells[25–28], but recent evidence indicates that it also promotes cell plasticity in non-malignant cells of multiple tissue origins, including macrophages[29–31].

Our data suggest that maintenance of homeostatic levels of ZEB1 in macrophages has an anti-atherogenic role by limiting the intracellular accumulation of lipids. ZEB1 expression was required for the transport of cholesterol through the endocytic pathway and its eventual efflux out of the macrophage. *ZEB1* mRNA is targeted by several microRNA networks, chiefly by members of the miR200 cluster, whose expression is also repressed by ZEB1 in a double negative feedback loop[25–27]. Interestingly, ZEB1 inhibits cell polarity in lung cancer cells, at least in part, by relieving miR200-mediated mRNA degradation of KIF13A and AP1S2, thus accelerating the endocytic transport of extracellular plasma membrane-bound proteins[69]. Likewise, repression of miR200 and TBC1D2b in lung cancer cells by ZEB1 complex induces E-cadherin endocytosis and endosomal degradation[70]. It remains to be elucidated whether these contrasting roles of ZEB1 in the intracellular trafficking of cholesterol versus plasma membrane-bound proteins depend on the cell type or the cargo.

*Zeb1*[ΔM] macrophages exhibited lower expression of several targets of the AMPK/LXRα signaling pathway and forced activation of AMPK in these macrophages reduced their enhanced lipid accumulation and restored *Nr1h3* and *Ppargc1a* expression to their same levels as control *Zeb1*[WT] macrophages. Conversely, inhibition of AMPK increased lipid accumulation in *Zeb1*[WT] macrophages but had no effect in *Zeb1*[ΔM] macrophages. AMPK signaling inhibits fatty acid synthesis and reduces cholesterol levels by inhibiting the expression and/or activity of SREBP1, FASN and the HMG-CoA reductase[71]. We found that the livers of *Zeb1*[ΔM]/*Apoe*[KO] mice exhibited higher levels of SREBP1c than *Zeb1*[WT]/*Apoe*[KO] counterparts. AMPK has dual roles in atherogenesis[72,73] and in the regulation of an EMT phenotype in cancer cells[26,27], promoting or inhibiting these processes through different mechanisms. These diverging effects of AMPK likely reflect the regulation of different downstream targets at different stages. In cancer cells, ZEB1 is not only activated by most signaling pathways, but its expression is also essential for enabling the complete functionality of these pathways[26,27]. For instance, ZEB1 downregulation inhibits Wnt signaling and many of its target genes[74]. Interestingly, AMPK activates the Wnt pathway and, just as for ZEB1, downregulation of Wnt signaling increases foam cell and atherogenesis by interfering with intracellular cholesterol traffic and lysosomal function[75].

Our data showed that ZEB1 deletion increases intracellular lipid content in macrophages, suggesting that homeostatic levels of ZEB1 in macrophages limit lipid overloading. As a regulator of cell plasticity, ZEB1 plays pleiotropic, often opposing functions, contingent upon the cell context. ZEB1 can activate or repress the expression of its target genes depending on the co-factors it recruits, which is determined by the particular gene, cell type and cell status[25–27,76,77]. Thus, ZEB1 overexpression increases lipid content in a pre-adipocyte cell line[78] and ZEB1 and other EMT-inducing transcription factors can trigger the transdifferentiation of proliferating cancer cells into post-mitotic lipid-laden adipocytes[79]. It is still unclear whether these diverging roles of ZEB1 in regulating lipid metabolism in macrophages, adipocytes, and cancer cells are mediated by distinct or overlapping mechanisms.

We found that ZEB1 expression in macrophages not only contributed to limiting inflammation within the atherosclerotic plaque and

arterial wall, but also dampened systemic inflammation. Atherosclerosis is a chronic inflammatory disease of the arterial wall, initiated by the infiltration of lipids and macrophages into the arterial intima[13]. oxLDL activates endothelial cells and macrophages to produce pro-inflammatory mediators, and the accumulation of cholesterol crystals within macrophages triggers further pro-inflammatory signaling[12,13,21,22]. We found that *Zeb1*[ΔM] macrophages accumulated more lipids, including in the form of cholesterol crystals, than *Zeb1*[WT] counterparts, thus contributing to higher immune infiltration of *Zeb1*[ΔM]/*Apoe*[KO] plaques. At the same time, atherosclerosis, including subclinical atherosclerosis, is associated with systemic inflammation[24,80,81]. Our data indicated that deletion of *Zeb1* in myeloid cells was sufficient to trigger several parameters of systemic inflammation in *Zeb1*[ΔM]/*Apoe*[KO] mice. For instance, compared to *Zeb1*[WT]/*Apoe*[KO] mice, *Zeb1*[ΔM]/*Apoe*[KO] mice had higher body weight, greater steatosis, higher WAT immune infiltration, higher serum levels of pro-inflammatory markers (IL1β, IL2, IL12, leptin, CCL2), higher cholesterol and LDL, and lower HDL and glucose tolerance.

ZEB1 functions are determined by fine thresholds of its expression levels. A partial downregulation of *Zeb1* in several cell types (e.g., cancer cells, tumor-associated macrophages, myofibers, adult muscle stem cells) is sufficient to block its homeostatic functions[29,30,34–38]. Heterozygous deletion of *Zeb1* in cancer mouse models blocks ZEB1's tumor-promoting functions[29,34,35,37,38] and further downregulation of ZEB1 does not have any additional effect on tumor formation but can prevent metastasis[34]. We found that exogenously induced expression of ZEB1 in *Zeb1*[ΔM] macrophages reduced intracellular lipid accumulation, increased cholesterol efflux and reduced the atherosclerosis plaque. However, overexpression of ZEB1 in *Zeb1*[WT] macrophages beyond homeostatic levels did not have any functional effect. These results suggest that in wild-type macrophages, ZEB1 is already bound to the regulatory regions of all of its target genes and additional copies of ZEB1 do not have additional effects. In human endarterectomies, we found a correlation between lower expression of *ZEB1* and an increased likelihood of IPH, plaque instability, and the occurrence of cardiovascular events. These patients with low levels of ZEB1 in their atherosclerotic plaques and a poorer prognosis—but probably not those with high ZEB1 levels and a better prognosis—would potentially benefit from therapeutic approaches aimed at increasing ZEB1 expression in macrophages.

ZEB1 is not only expressed but is essential for the homeostatic functions of various non-epithelial cells. Downregulation of ZEB1 in some of these non-epithelial cells is linked to the development or exacerbation of various physiopathological conditions and diseases (e.g., corneal dystrophy, muscle atrophy, muscular dystrophy, osteoporosis, post-ischemic brain damage and, shown here, atherosclerosis plaque formation) and transient or stably ZEB1 overexpression has been used to treat these conditions in preclinical models[30,36,82–84]. Because of ZEB1's role in tumor initiation and progression, the overexpression of ZEB1 for therapeutic purposes naturally raises concerns. However, it should be noted that our approach to upregulating ZEB1 differs from some previous strategies because we specifically directed ZEB1 expression to differentiated myeloid cells using nanoparticles where *Zeb1* cDNA is driven by the *LysM* promoter, thus preventing ZEB1 overexpression in any non-myeloid cell. Nevertheless, since the role of ZEB1 in macrophages is still emerging, future research is needed to determine whether prolonged ZEB1 overexpression in myeloid cells

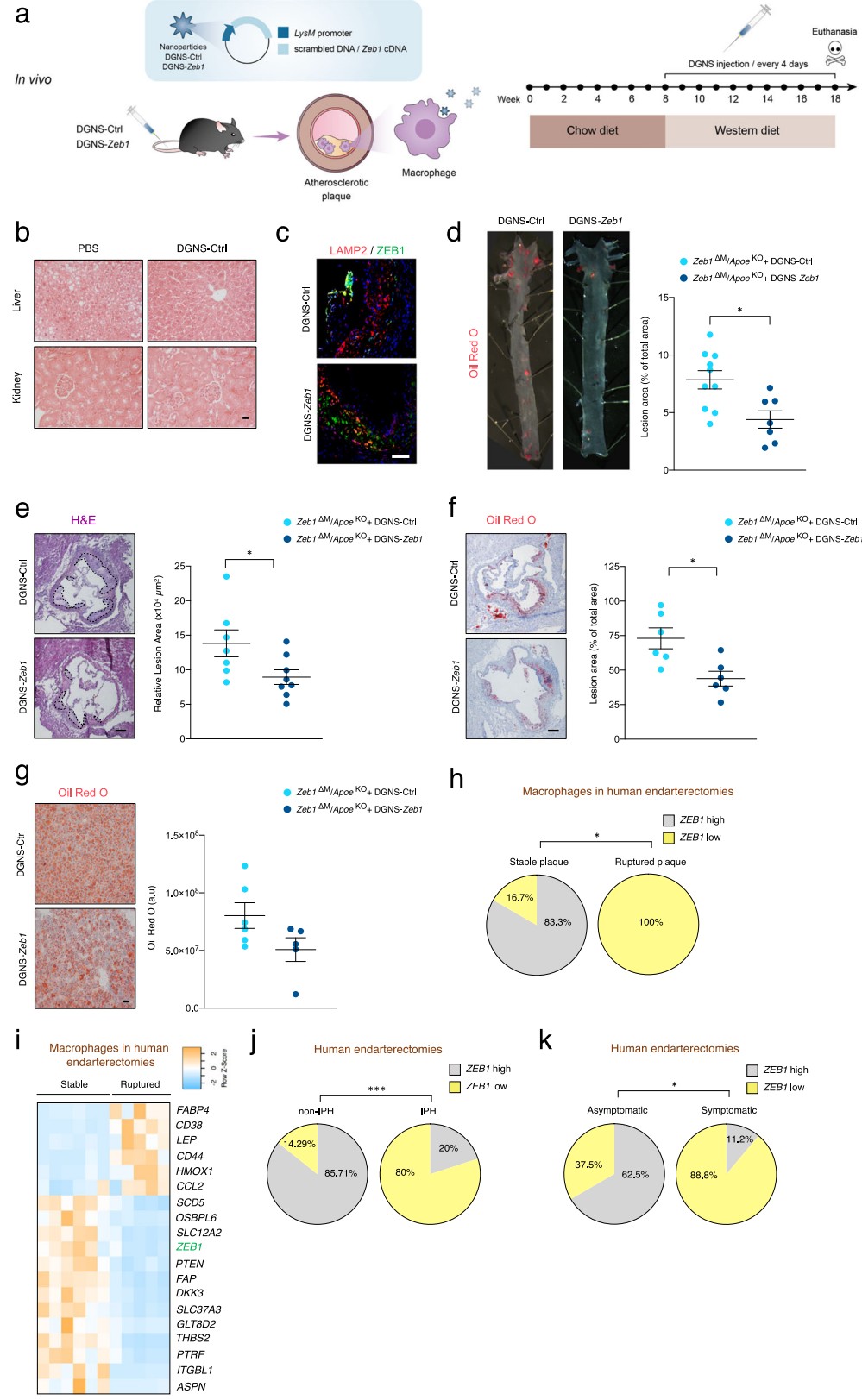

using nanoparticles can have deleterious effects and/or potentially contribute to cancer development.

Collectively, the findings presented here establish ZEB1 as an anti-atherogenic factor in macrophages, suggesting that increasing its expression could be a potential therapeutic strategy for athero-sclerosis in patients with low ZEB1 levels.

## Methods

### Research ethics statement

The research in this study followed all relevant ethical guidelines. Information on the ethical approvals related to the use of human samples and mouse models is detailed in the sections below.

**Fig. 6 | In vivo use of macrophage-targeted nanoparticles expressing ZEB1 reduces atherosclerotic plaque formation. a** Schematic and timeline of the injection of 50 μg/kg of DGNS-Ctrl or DGNS-*Zeb1* in *Zeb1*^ΔM^/*Apoe*^KO^ mice. **b** As in **a**, H&E staining of sections of liver and kidney from *Zeb1*^ΔM^/*Apoe*^KO^ mice injected with DGNS-Ctrl. Scale bar: 20 μm. ($n = 2,3$) **c** Representative pictures of aortic sinus sections *Zeb1*^ΔM^/*Apoe*^KO^ mice. stained for ZEB1 (HPA027524, 1/500) and LAMP2/MAC3 (clone M3/84, 1/50). ($n = 3$). Scale bar: 50 μm. **d** Representative images of the *en face* ORO staining of aorta of *Zeb1*^ΔM^/*Apoe*^KO^ mice at the end of the protocol as in (**a**) and quantification of the plaque area by ImageJ ($n = 10,7$). **e** Representative pictures of H&E staining of aortic sinus sections of *Zeb1*^ΔM^/*Apoe*^KO^ mice and quantification of the plaque area ($n = 7,8$). Scale bar: 200 μm. **f** As in **d**, but staining for ORO. Scale bar: 200 μm ($n = 6$). **g** As in **f**, but in liver sections ($n = 6,5$). **h** Data from eleven plaques (7 males; 4 females), six stable and five rupture, from the endarterectomies of patients in GSE41571 were assessed for *ZEB1* mRNA expression, divided into two cohorts by the median of *ZEB1* expression (*ZEB1*^high^ versus *ZEB1*^low^), and examined for their association with stable or ruptured plaques. **i** As in **h**, heatmap of genes associated with plaque stability and rupture in stable and ruptured plaques. **j** Analyses of plaques from the endarterectomies of 39 male patients in GSE163154[68]—14 without intra-plaque hemorrhage (non-IPH) and 25 with plaque hemorrhage (IPH)—were assessed for *ZEB1* mRNA expression divided in two cohorts by the median of *ZEB1* expression (*ZEB1*^high^ versus *ZEB1*^low^), and examined for their association with IPH. **k** The endarterectomies from 24 patients (21 males; 3 females) were assessed for *ZEB1* mRNA levels by qRT-PCR and segregated in two cohorts depending on whether they expressed *ZEB1* above (*ZEB1*^high^, 11 patients) or below (*ZEB1*^low^, 13 patients) the median. The clinical history of patients in both cohorts was analyzed to determine whether they had (symptomatic) or not (asymptomatic) a cardiovascular event. Graphs represent mean values ± SEM with two-tailed unpaired Mann-Whitney (**d, e, f**) or Fisher's exact test (**h, j, k**). $p \leq 0.001$ (***), $p \leq 0.01$ (**) or $p \leq 0.05$ (*) levels, or non-significant (ns) for values of $p > 0.05$. The raw data, along with *p* values from statistical analyses are included in the Source Data file.

## Plasmids

Expression plasmids harboring a scrambled non-coding cDNA and mouse *Zeb1* cDNA expression plasmids under the *Lyz2* (*LysM*) promoter custom synthesized by VectorBuilder (Guangzhou, China).

## Human samples

The tissue and RNA samples from human subjects used in the study were obtained with the approval of the local institutional review boards (Clinical Ethics Research Committee at Hospital of Bellvitge and Hospital Clinic of Barcelona) (protocols 14/04 and 23/0249, respectively). Samples were obtained with the written informed consent of patients, and adhered to the principles of the Helsinki Declaration. Samples of atheroma plaques originate from 24 patients (21 males; 3 females) subjected to programmed endarterectomy in two cohorts: symptomatic or asymptomatic dependent on whether or not they had suffered a cerebrovascular accident, grade IV to VI as previously reported (Supplementary Table S1)[85].

## Generation of the conditional *Zeb1*^WT^ and *Zeb1*^ΔM^ mice

The development of the conditional *Zeb1* flox allele mouse (*Zeb1*^fl/fl^, which is herein designated as *Zeb1*^WT^) was carried out at the Transgenesis Unit, which is jointly affiliated with the National Biotechnology Center (CNB) and the Severo Ochoa's Molecular Biology Center-Autonomous University of Madrid (CBMSO-UAM) (Madrid, Spain), both centers belonging to the Spanish National Research Council (CSIC). The procedure for creating the *Zeb1*^WT^ mouse is detailed in Supplementary Fig. S1d. In summary, the Breaking-Cas online platform[86] (https://bioinfogp.cnb.csic.es/tools/breakingcas/) was employed to design two gRNAs (sgRNA5'and sgRNA 3') (see Supplementary Table S2) for inducing double-strand breaks (DSBs) flanking exon 6 in the *Zeb1* gene. Two oligonucleotides for each targeting site (Supplementary Table S2) were annealed and ligated into the pX330-U6-Chimeric_BB-CBh-hSpCas9 vector (Addgene plasmid # 42230; RRID: Addgene_42230)[87], a gift from Dr. F Zhang (Massachusetts Institute of Technology, Cambridge, MA, USA). Then, T7 promoter sequence was incorporated to the sgRNA template for subsequent amplification by PCR and in vitro transcription (IVT). Subsequently, two ssDNA oligos harboring the corresponding LoxP site and a restriction enzyme flanked by two 60 bp homology arms, which correspond to the sequence surrounding each Cas9 cut site, were designed and purchased from Sigma-Aldrich (Merck KgaA, St. Louis, MO, USA) (see Supplementary Table S2) (Merck KGaA, Darmstadt, Germany). The cytoplasm of B6CBAF2 zygotes was then injected with a mixture of in vitro transcribed RNA Cas9 (Sigma-Aldrich), two sgRNAs, and two ssDNAs(ssDNA 5' + ssDNA 3'). The concentrations of RNAs injected were: 100 ng/μl for Cas9, 50 ng/μl for each sgRNA 5'- and sgRNA 3'-, and 100 ng/μl for each ssDNAs. Zygotes that survived the injections were transferred into the oviducts of pseudopregnant foster mothers for development to term. The resulting progeny was then crossed five times with wild-type C57BL6/J mice to generate the *Zeb1*^fl/+^ mice (mice expressing one floxed allele and one wild-type allele). The presence of correct LoxP sequences was analyzed by DNA sequencing (Supplementary Table S2). The *Zeb1*^fl/fl^ (*Zeb1*^WT^) mouse (B6.B6CBA-*Zeb1*em1/cnbbm) was then crossed with a mouse carrying the Cre recombinase selectively in myeloid cells under the control of the endogenous lysozyme 2 (*Lyz2*, also referred as *LysM*) promoter/enhancer (official name: B6.129P2-*Lyz2*^tm1(cre)Ifo^/J)[88], (The Jackson Labs, Bar Harbor, ME, USA), to generate the myeloid conditional *Zeb1* knockout (*Zeb1*^fl/fl^/*LysM*^Cre^, referred in the manuscript as *Zeb1*^ΔM^) mice. The floxed and Cre sequences in experimental mice were genotyped by PCR amplification of genomic DNA extracted from tail samples. In Fig. 1b, c as well as in Supplementary Fig. S1c, we used a constitutive heterozygous mouse [*Zeb1*(+/−)], which was obtained from Dr. Douglas S. Darling (University of Louisville, KY, USA) and Dr. Y. Higashi (Institute for Development Research, Kasugai-shi, Aichi, Japan)[33].

## Atherosclerosis mouse model and experimental mice used in the study

The use of mice in the study followed the guidelines established by the University of Barcelona School of Medicine and Health Sciences' Animal Experimental Committee and by the Generalitat de Catalonia which reviewed and approved it under references 258/17 and 358/18, respectively. Wild-type [referred to as *Zeb1*(+/+)], *Zeb1*(+/−), *Zeb1*^WT^, and *Zeb1*^ΔM^ mice were crossed with *Apoe*(−/−) (referred hereafter to as *Apoe*^KO^) mice (B6.129P2-Apoe^tm1Unc^/J) from The Jackson Laboratory (Strain #:002052)[39] to generate the four experimental mouse models used in the study, namely *Zeb1*(+/+)/*Apoe*^KO^, *Zeb1*(+/−)/*Apoe*^KO^, *Zeb1*^WT^/*Apoe*^KO^ and *Zeb1*^ΔM^/*Apoe*^KO^ mice. From birth to the age of 8 weeks, male mice are fed ad libitum with the Chow diet (RM1-P, SDS, Dietex, Argenteuil, France) (see schematics in Figs. 1a, d, and 6a). During the following 10 weeks (week 8 to 18) in the case of *Zeb1*^WT^/*Apoe*^KO^ and *Zeb1*^ΔM^/*Apoe*^KO^ mice or the following 12 weeks (week 8 to 20) in the case of *Zeb1*(+/+)/*Apoe*^KO^ and *Zeb1*(+/−)/*Apoe*^KO^ mice, male mice were fed ad libitum with Western diet food (high fat/high cholesterol, 21% butter fat and 0.21% cholesterol. E15721-34, corresponding to TD.88137), which was purchased from Ssniff Special Diets GmbH (Soest, Germany). The study utilized only male mice because, despite Western-fed *Apoe*^KO^ female mice developing larger atherosclerotic plaques, males in atherosclerosis mouse models fed with Western diet exhibit higher levels of serum lipoproteins and triglycerides, gain more weight and, more importantly, display higher levels of inflammatory markers[89–92]. In fact, the inflammatory state in atherosclerosis mouse models is considered more relevant for predicting sex-specific outcomes in humans than plaque size[1,93]. All the mice in the study were housed in a temperature-controlled barrier room maintained at 21–22 °C with a 12-h light/dark cycle. Mice had access to food and water ad libitum and were euthanized by carbon dioxide inhalation.

## Mouse aortic root preparation and tissue staining

At the end of the feeding protocol, mice were sacrificed and their hearts and aortas were perfused with PBS and collected for further analysis. Aortas were fixed in 4% paraformaldehyde (PFA) aqueous solution (Electron Microscopy Sciences, Hatfield, PA, USA) at 4 °C. Hearts including the aortic root, livers, and kidneys were embedded in Tissue-Tek® OCT™ compound (Sakura Finetek, The Netherlands) and stored at −80 °C. Skeletal gastrocnemius muscle samples were frozen in 2-methylbutane under nitrogen liquid and stored at −80 °C. For *en face* staining, complete aortas were washed and the surrounding adipose tissue was eliminated. After washing once with PBS, twice with methanol 78%, aortas were stained for 1 h with 0.5% Oil Red O (ORO) ($C_{26}H_{24}N_4O$) (Sigma-Aldrich, Merck KGaA) solution in 25 mL 100% methanol with 10 ml 1 M NaOH and then washed twice with methanol 78%, sectioned longitudinally and pinned flat. The lesion area was quantified as the ratio between the ORO$^+$ area and total aorta area. To analyze the cross-sectional lesion area, hearts were cut into 7 μm sections through the aortic sinus. For each mouse, three sections were stained. Hematoxylin and eosin (H&E) staining of tissue sections (aortic sinus, liver, kidney, skeletal muscle) was carried out by equilibrating tissues for 5 min at RT, fixed in 4% PFA, incubated in Harris Hematoxylin Modified Solution (Panreac Quimica SLU, Panreac-AppliChem GmbH, Barcelona, Spain) for 3 min, washed once in ethanol/LiCl, 5 min incubation with Wright's eosin methylene blue (Sigma-Aldrich), and four in absolute ethanol. Sirius Red (Sigma-Aldrich) staining was carried in 7 μM thickness sections fixed with PFA 4% and immersed in Sirius Red 0.02% in 1.3% saturated picric acid solution for 1 h. Masson's trichrome staining was performed at IDIBAPS' Tumor and Tissue Bank according to standard protocols. Sections were stained with Hansen's iron hematoxylin for 5 min and rinsed under running tap water for 5 min. The sections were then stained for 10 min with Biebrich scarlet-acid fuchsin. After rising with water, the slices were treated for 10 min with phosphomolybdic acid before being transferred directly to aniline blue solution for 10 min. The collagen fibers displayed a blue color after staining. Lipid content in tissue sections of the aorta, liver, kidneys, and skeletal muscle was assessed by ORO. All tissues were cut in sections of 12 μm thickness in a Leica CM1950 cryostat (Leica Biosystems, Heidelberger, Germany). Tissue sections stored at −80 °C were equilibrated at room temperature for 5 min before the staining process. Aorta, liver and kidney tissue sections were fixed with 4% PFA. Skeletal muscle sections were stained without fixation. The ORO working solution consists of 60% ORO stock solution (0.5% ORO in isopropanol, Sigma-Aldrich) and 40% distilled water. The mixture was allowed to stand for 10 min at RT before being filtered to remove any precipitates. Sections were then incubated with the ORO working solution for 15 min at RT, followed by a wash under running tap water. Images were captured on a bright-field Olympus BX41TF microscope and Cell Sens software (Olympus America Inc., Melville, NY, USA) and quantified with ImageJ software (NIH, Bethesda, MD, USA) (https://imagej.nih.gov/).

## In situ atherosclerotic plaque efferocytosis assay

Aortic sinus sections from *Zeb1*$^{WT}$/*Apoe*$^{KO}$ and *Zeb1*$^{ΔM}$/*Apoe*$^{KO}$ mice stored at −80 °C were equilibrated for 10 min at room temperature and fixed with 4% PFA. Sections were subsequently stained with TUNEL (terminal deoxynucleotidyl transferase (TdT) dUTP Nick-End Labeling) using an Alexa Fluor™ 488 Click-iT™ Plus TUNEL Assay kit (reference C10617, Invitrogen™, Thermo Fisher Scientific, Waltham, MA, USA) according to the manufacturer's instructions. Images were captured using a Zeiss Axiovert 200 inverted microscope (Zeiss, Berkochen, Germany). For efferocytosis quantification, the number of apoptotic TUNEL$^+$ cells that co-localized with macrophages (LAMP2/MAC-3$^+$) ("associated", indicative of efferocytosis) and the number of apoptotic TUNEL$^+$ cells that were not surrounded by or in contact with macrophages ("free", not undergoing efferocytosis) were assessed using

ImageJ software. Efferocytosis efficiently is then calculated as the ratio of associated/free apoptotic TUNEL$^+$ cells per aortic sinus section.

## LDL isolation and oxidation

Whole human blood was obtained from healthy donors at the Catalan Blood and Tissue Bank. To separate human plasma, the blood was centrifuged at 2000 × g for 10 min. LDL was isolated by ultracentrifugation in KBr density gradient mixing with KBr to a density of 1.21 g/l and subsequently adding 1.063 g/l, 1.019 g/l and 1.006 g/l density solutions. The resulting LDL was dialyzed against PBS in a high retention dialysis tubing (Sigma-Aldrich) for 48 h and concentrated using poly-ethylene glycol 20,000 MW (Thermo Fisher Scientific). Oxidation of LDL was performed by incubation with 5 μmol/L of $CuSO_4$ for 20 h at 37 °C in an oxygenated atmosphere. The reaction was stopped by adding EDTA to a final concentration of 0.24 mM. The LDL solution was then dialyzed for 48 h at 4 °C to remove excess copper.

## Isolation and culture of peritoneal macrophages

Peritoneal macrophages were isolated from 6- to 8-week old male mice. After euthanasia, the peritoneal cavity of the mice was washed twice with 6 ml of ice-cold PBS supplemented with 3% PBS and peritoneal lavage was centrifuged at 400 × g for 10 min at 4 °C to obtain the cells. Erythrocytes were osmotically lysed with red blood cell lysis buffer (Sigma-Aldrich), followed by a wash with PBS and resuspension in PBS with 3% of PBS. Peritoneal macrophages were then isolated by flow cytometry, targeting CD45$^+$, CD11b$^+$, F4/80$^+$ cells (see Supplementary Fig. S7 for the gating strategy). Isolated cells were then processed for mRNA analysis. Where indicated, macrophages were enriched out of total peritoneal cells by plating 5 × 10$^5$ peritoneal cells on tissue culture dish plates for 30 min and then washed to remove non-attached cells. Wherever indicated macrophages were cultured in RPMI 1640 medium containing 10% FBS and incubated for 24 h with 50 μg/mL ox-LDL and/or 100 μM A-769662 (MedChemExpress, Monmouth Junction, NJ, USA) or treated with 15 μM of Compound C (6-[4-(2-Piperidin-1-ylethoxy) phenyl]−3-pyridin-4-ylpyrazolo [1,5-a]pyrimidine) (Sigma-Aldrich, Merck KGaA) for 3 h and then stimulated with 50 μg/mL ox-LDL for 20 h. Cells were fixed using 4% PFA for 1 h at RT. Neutral lipids were then stained with 1 μg/mL BODIPY® 493/503 (4,4-Difluoro-1,3,5,7,8-Pentamethyl-4-Bora-3a,4a-Diaza-s-Indacene) (Invitrogen™, Thermo Fisher Scientific) and free cholesterol was stained with 0.05 mg/ml of filipin (Sigma-Aldrich). Fluorescence was evaluated in an inverted fluorescence microscope, either a Leica AF6000 (Leica Biosystems) or a Zeiss Axiovert 100 (Zeiss).

## Isolation of aortic plaque macrophages

Infiltrating macrophages in the whole aorta of *Zeb1*$^{WT}$/*Apoe*$^{KO}$ and *Zeb1*$^{ΔM}$/*Apoe*$^{KO}$ mice at the end of the Western diet were isolated as follows. Briefly, whole aortas were harvested and separated from perivascular adipose tissue, then digested for 40 min at 37 °C in RPMI containing 450 U/ml Collagenase I, 125 U/ml Collagenase XI and 60 U/ml Hyaluronidase. After digestion, cells were blocked with mouse gamma globulin and stained with CD45-PerCP, CD11b-PE, F4/80-APC and eBioscience™ Fixable Viability Dye eFluor™ 450 (Thermo Fisher Scientific) at 4 °C for 40 min. Viable CD45$^+$/CD11b$^+$ / F4/80$^+$ cells were sorted in a FACS Aria II (BD Biosciences, San Jose, CA, USA) (the gating strategy is shown in Supplementary Fig. 7), recovered in lysis buffer [0.2% Triton X-100 (Sigma-Aldrich) in water with 100 U of RNAse inhibitor (Life technologies, Thermo Fisher Scientific, Carlsbad, CA, USA)], and frozen at −80 °C until further processing for low-input RNA sequencing. To characterize macrophage subtypes, cells were labeled with antibodies against CD45-PerCP, F4/80-Alexa488, CD11b-PE, CD9-APC, and CD86-BV421 at 4 °C for 45 min and subsequently analyzed using an LSRFortessa™ cell analyzer (BD Biosciences). The identity and source of antibodies are listed in Supplementary Table S3.

## Isolation of liver macrophages

Liver macrophages from $Zeb1^{WT}/Apoe^{KO}$ and $Zeb1^{ΔM}/Apoe^{KO}$ mice fed with the Western diet protocol were isolated as follows. The mouse liver was perfused with calcium-free HBSS until the organ began to swell. Subsequently, the liver was isolated, and the gallbladder was removed. The liver was then cut into pieces and digested in collagenase IV and DNAse digestion buffer for 30 min at 37 °C. Cell pellet was resuspended in a 33% Percoll solution and centrifuged at $1000 \times g$ for 20 min at RT with 1 of acceleration and no brake. Cell pellet that contains immune cells was stained with CD45-PerCP, CD11b-PE, F4/80-APC and DAPI for flow cytometry. CD45$^+$ CD11b$^+$ and F4/80$^+$ (triple-positive) cells were sorted in a FACS Aria II (BD Biosciences) (the gating strategy is shown in Supplementary Fig. S7).

## Isolation of white adipose tissue macrophages

Epididymal fat pads were dissected from both sides of $Zeb1^{WT}/Apoe^{KO}$ and $Zeb1^{ΔM}/Apoe^{KO}$ mice fed with the Western diet protocol. Fat pads were washed with PBS, cut into pieces and incubated with collagenase II (Sigma-Aldrich) for 30 min at 37 °C under constant shaking. After digestion, cells were spun down and stained with antibodies against CD45-PerCP, CD11b-PE and F4/80-APC at 4 °C for 40 min. The identity and source of antibodies are listed in Supplementary Table S3. CD45$^+$ CD11b$^+$ and F4/80$^+$ (triple-positive) cells were sorted in a FACS Aria II (BD Biosciences) (the gating strategy is shown in Supplementary Fig. S7).

## Determination of peripheral blood cell populations

Total blood from the tail of mice fed on the Western diet was collected in EDTA tubes. Counts for the different blood cell populations were quantified in a BC-2800Vet automatic Hematology Analyzer (Shenzhen Mindray Bio-Medical Electronics Co., Ltd., Shenzhen, PR China).

## Assessment of cell surface protein expression analysis and cell sorting by flow cytometry (FACS)

Cells were blocked for Fc receptors with mouse gamma globulin for 30 min at 4 °C (Jackson ImmunoResearch Europe). Then, cells were incubated with fluorochrome-labeled antibodies in PBS with 3% FBS at 4 °C for 45 min. Analysis of protein surface expression was assessed in a BD FACSCanto™ III analyzer (BD Biosciences) or cells were sorted for specific subpopulations in FACS Aria™ II cell sorter (BD Biosciences) for use in selected experiments.

## Assessment of protein expression by immunofluorescence

Mouse peritoneal macrophages at 6–8 weeks old were pelleted, plated on glass coverslips (Fisher Scientific), and incubated for 1 h at 37 °C with 50 μg/mL of oxLDL. To assess protein expression by immunofluorescence, cells were first fixed for 1 h at room temperature with Bouin's solution (Sigma-Aldrich, Merck) and permeabilized for 30 min at room temperature with 0.1% saponin (Sigma-Aldrich) with 3% BSA fraction V (Sigma-Aldrich). Cells were then washed thrice with 4 °C PBS and incubated for 1.5 h at room temperature with the corresponding primary antibodies (Supplementary Table S3): LAMP1 (clone H4A3, 1/500), NPC2 (19888-1-AP, 1/100). After washing with 4 °C PBS, cells were incubated for 1 h at room temperature with their corresponding fluorochrome-conjugated secondary antibodies (Supplementary Table S3): Alexa Fluor 488 donkey anti-mouse (1/200 dilution) and Rhodamine X donkey anti-rabbit (1/200). Cells were then counterstained with ProLong™ Gold Antifade Mountant with DAPI (P36931, Invitrogen™, Thermo Fisher Scientific), mounted on Menzel™ Gläser Superfrost Plus (Epredia Inc., Kalamazoo, MI, USA), and examined in a Zeiss LSM880 confocal microscope (Zeiss) equipped with 63×/1.4 Oil objective.

## Assessment of protein expression by western blot

Livers were collected at the end of the Western diet feeding protocol, frozen in liquid nitrogen and digested in small sections. Then, the homogenate was lysed with radio immunoprecipitation assay (RIPA) buffer (50 mM Tris pH 8.0, 150 mM NaCl, 2 mM EDTA, 1% NP-40, 0.5% deoxycholic sodium acid, 0.5% SDS) containing protease inhibitors (1 mg/ml leupeptin, 1 mg/ml pepstatin, 1 mg/ml apoptin, 100 mM PSMF) and with subsequent sonication 3 s by 3 pulses at 18% of amplitude. Protein concentration was quantified using the RC DC™ Protein Assay kit (Bio-Rad, Hercules, CA, USA). The WB loading Buffer (0.25 M Tris pH 6.8, 1.2% SDS, 0.06% bromophenol blue, 45% glycerol plus 6 mM freshly made DTT) was added to samples prior to boiling them at 95 °C for 5 min and then loaded onto a Novex™ WedgeWell™ of 4–20% gel (Invitrogen™, Thermo Fisher Scientific). Polyvinylidene difluoride (PVDF) membranes (Immobilon®-P; Millipore, Burlington, MA, USA) were activated with methanol and then gels were transferred onto the membranes. To block for non-specific binding, membranes were incubated with non-fat dry milk 5% (Panreac Quimica SLU) and immunoblotted overnight at 4 °C with the corresponding primary antibodies listed in Supplementary Table S3. The chemiluminescent reaction was detected with Clarity ECL Western Blotting Substrate (Bio-Rad). Quantification of expression with respect to GAPDH was assessed with ImageJ software. Full uncropped gel images are shown in the Source Data File, with the target protein enclosed by a dashed black box.

## RNA extraction and assessment of gene expression by quantitative real-time PCR

Total RNA was extracted with TRIzol® (Life Technologies, Thermo Fisher Scientific) and reverse-transcribed with oligo-dT using High-Capacity cDNA Reverse Transcription Kit (Life Technologies, Thermo Fisher Scientific). mRNA levels were then determined by quantitative real-time PCR (qRT–PCR) at 60 °C using GoTaq® qPCR Master Sybr Green Mix (Promega Corp., Madison, WI, USA) in a LightCycler ® 96 real-time PCR apparatus (Roche, Rotkreuz, Switzerland). Results were analyzed using Opticon Monitor 3.1.32 software (Bio-Rad) by ΔΔCt method and normalizing values with respect to $Rpl19$ (mouse samples) and $GAPDH$ (human samples) as reference genes with similar results. qRT-PCR data shown are the average of a minimum of three mice of each genotype run in triplicate. DNA primers used in qRT-PCR were purchased from Sigma-Aldrich, and their sequences are described in Supplementary Table S4. Throughout the study, the nomenclature for human and mouse genes used adheres to the *HUGO Gene Nomenclature Committee* (http://www.genenames.org/) and *Mouse Genome Informatics* (http://www.informatics.jax.org/) nomenclatures, respectively.

## RNA sequencing and data analysis

Macrophages from the aortic sinus of $Zeb1^{WT}/Apoe^{KO}$ and $Zeb1^{ΔM}/Apoe^{KO}$ male mice, subjected to the Western diet protocol, were sorted by FACS as described above. Libraries were prepared in triplicate. Macrophages from three mice samples were mixed from each genotype and condition. cDNA and libraries were prepared using SMART-Seq® v4 Ultra® Low Input RNA Kit for Sequencing (634898, Takara Bio., Kusatsu, Japan). The cDNA and libraries quality were examined on an Agilent 2100 Bioanalyzer. Libraries were sequenced on HiSeq 4000 (Illumina Inc.) in a paired-end mode with a read length of 2 × 75 bp using TruSeq SBS Kit v3-HS (Illumina, San Diego, CA, USA). Over 30 million paired-end reads were generated for each sample in a fraction of a sequencing flow cell lane, following the manufacturer's protocol. Image analysis, base calling, and quality scoring of the run were processed using the manufacturer's software Real-Time Analysis (RTA 1.13.48) and followed by the generation of FASTQ sequence files by Illumina's CASAVA 1.8.2 software. Reads were mapped against the mouse reference genome (GRCm38) with STAR (https://code.google.com/archive/p/rna-star/)[94] using the ENCODE parameters for long RNA. Gene quantifications were performed with RSEM (https://deweylab.github.io/RSEM/)[95] using default options and mouse GEN-CODE annotation version 25 (https://www.gencodegenes.org/mouse/

release_M25.html). Normalization and differential expression analyses were done with DESeq2 (https://bioconductor.org/packages/release/bioc/html/DESeq2.html)[96] using default options. Heatmap with the regularized log-transformed expression of the most variable genes was performed with the 'Pheatmap' R package and the option scale = "row". Gene network analysis of the RNAseq was conducted with NetworkAnalyst 3.0 platform (https://www.networkanalyst.ca/)[97] (Supplementary Fig. S4a). The RNAseq data have been submitted to the GEO database and assigned accession number GSE206477. We also analyzed two published RNAseq datasets of human endarterectomies, namely GSE41571 (7 males, 4 females[45]) (https://www.ncbi.nlm.nih.gov/geo/query/acc.cgi?acc=GSE41571) and GSE163154 (39 males[68]).

### Analysis of single-cell RNAseq datasets

We conducted an integrated analysis of two single-cell RNA sequencing (scRNAseq) datasets from $Apoe^{KO}$ mice subjected to a high-fat diet (GSE116240[3], and GSE149070, https://www.ncbi.nlm.nih.gov/geo/query/acc.cgi?acc=GSE116240[4]) as well as three scRNAseq datasets from atherosclerotic coronary arteries in humans (ref. 5 and https://figshare.com/s/c00d88b1b25ef0c5c788; and GSE131778[6]). Twelve scRNA-seq datasets of mouse and 3 scRNA-seq datasets of human[8] were integrated separately based on the species to do the cell type annotation after being pre-processed individually. Violin plots were subsequently generated based on the human dataset and a subset of cells from two $Apoe^{KO}$ mice high fat diet datasets. All data were processed mainly in R package Seurat (version 4.3.0.1) with code included in ref. 8.

### Ultrastructure of the atherosclerotic plaque by electron transmission microscopy

For transmission electron microscopy analysis, the aortas of $Zeb1^{WT}$/$Apoe^{KO}$, and $Zeb1^{ΔM}$/$Apoe^{KO}$ mice at the end of the Western diet protocol were harvested, dissected from the perivascular adipose tissue, cut into 1 mm³ pieces and subjected to two rounds of fixation—the first round of 2 h and the second round of 72 h—in freshly prepared cold 3% glutaraldehyde solution in 0.1 M phosphate buffer. Post-fixation was performed with 1% OsO$_4$ for 90 min at 4 °C. Samples were then dehydrated, embedded in Spurr resin (Electron Microscopy Sciences, Hatfield, PA, USA), and sectioned using Leica ultramicrotome (Leica Microsystems, Wetzlar, Germany). Ultra-thin sections (50–70 nm) were stained with 2% uranyl acetate for 10 min and with a lead-staining solution for 5 min and observed using a JEOL JEM-1010 transmission electron microscope (JEOL Ltd., Akishima, Tokyo, Japan) coupled with an Orius SC1000 CCD camera (model 832) (Gatan Inc., Pleasanton, CA, USA) at the Unit of Electron Microscopy, Scientific and Technological Centers of the University of Barcelona, School of Medicine and Health Sciences (Barcelona, Spain).

### Ultrastructure of macrophages by electron transmission microscopy

Peritoneal cells from 6- to 8-week-old $Zeb1^{WT}$ and $Zeb1^{ΔM}$ male mice were isolated and exposed to oxLDL for 24 h. Macrophages were sorted by FACS and pooled from 6 mice of each genotype. Sorted macrophages were then washed in PBS and fixed for 1 h in 500 µl of 3% glutaraldehyde in 0.1 M phosphate buffer at room temperature, centrifuged, and resuspended in 500 µl of fresh ice-cold 3% glutaraldehyde 0.1 M phosphate buffer, and fixed for 72 h at 4 °C. Macrophages were then pelleted in Eppendorf tubes, washed in phosphate buffer, and incubated with 1% OsO4 for 90 min at 4 °C. Samples were then dehydrated, embedded in Spurr's Low Viscosity embedding mixture (Electron Microscopy Sciences), and sectioned using Leica ultramicrotome (Leica Microsystems). Ultra-thin sections (50–70 nm) were stained with 2% uranyl acetate for 10 min and with a lead-staining solution for 5 min and observed using a JEOL JEM-1010 transmission electron microscope (JEOL Ltd.) coupled with an Orius SC1000 CCD camera (model 832) (Gatan Inc.) in the Electron Microscopy Unit at the University of Barcelona School of Medicine and Health Sciences (Barcelona, Spain).

### Determination of lipids, albumin, total protein, and transaminases in serum

At the end of the Western diet feeding protocol, mice were subjected to an overnight fast, and blood was collected upon euthanasia and allowed to incubate at room temperature until coagulation occurred. Serum was separated by centrifugation at 1500 × $g$ for 10 min and 4 °C. Serum concentrations of total cholesterol, free cholesterol, and triglycerides were measured using commercial kits adapted for a COBAS® 6000 autoanalyzer (Roche). HDL cholesterol levels were measured in the supernatant obtained after the precipitation of serum VLDL/LDL lipoprotein particles with 0.44 mmol/L phosphotungstic acid (Merck KGaA) and 20 mmol/L magnesium chloride (Sigma-Aldrich). In parallel, the LDL fraction was isolated from the serum via sequential ultracentrifugation, using potassium bromide for density adjustment (1.019-1.063 g/mL), at 100,000 × $g$ for 24 h with an analytical fixed-angle rotor (Beckman Coulter, Fullerton, CA) and the cholesterol concentration measured in the LDL fraction using a commercial kit adapted for a COBAS® 6000 autoanalyzer. To examine the potential toxicity of dendrimer-graphene nanostar (DGNS), total blood from the tail of mice fed on the Western diet and injected with DGNS-Ctrl was collected in EDTA tubes. Serum albumin, total protein, alanine aminotransferase (ALT) and aspartate aminotransferase (AST) were measured using a BS-200E Chemistry Analyzer (Shenzhen Mindray Bio-Medical Electronics Co., Ltd).

### Determination of serum glucose and glucose tolerance test

Following 12 weeks under the Western diet feeding protocol, $Zeb1^{WT}$/$Apoe^{KO}$ and $Zeb1^{ΔM}$/$Apoe^{KO}$ mice were fasted for 16 h (allowed to drink water) and injected intraperitoneally with 2 g/kg body weight of glucose (Sigma-Aldrich). At 15, 30, 60, and 90 min after the administration of glucose, a drop of blood from the tail was applied to the tip of glucose strips (Nova Biomedical Corporation, Waltham, MA, USA) from where serum glucose levels were measured using a NovaPro glucose monitoring system (Nova Biomedical Corporation).

### Ex vivo cholesterol efflux capacity

Peritoneal macrophages obtained from 6- to 8-week-old male mice were plated on 96-well plates in RPMI 1640 medium and allowed to adhere for 30 min. Afterward, they were washed twice with PBS and cultured in RPMI 1640 medium for a minimum of 4 h. Subsequently, the cells were treated with 50 µg/mL of oxLDL and 1 µCi/mL [³H]cholesterol for 24 h. Then, cells were equilibrated in serum-free RPMI 1640 medium containing 0.2% (w/v) bovine serum albumin (BSA) for 16 h, followed by two washed in PBS and the incubation with 25 µg/mL lipid-free APOAI (Sigma-Aldrich) or 50 µg/mL HDL for 4 h. After efflux, medium was collected and centrifuged to remove any detached cells. Cells were washed in PBS and lysed in 0.2 M NaOH for 24 h. Medium and cell lysates were counted for radioactivity. Cholesterol efflux was calculated using the following formula:

$$\text{Cholesterol efflux (\%)} = 100 \times (^3\text{H-cholesterol in medium}/^3\text{H-cholesterol in medium} + {}^3\text{H-cholesterol in cells})$$

The cholesterol efflux of wells without acceptors was subtracted from those containing the acceptors. All conditions were measured by three technical replicates.

### pHrodo™ Green-LDL and DiI-oxLDL intracellular trafficking

To assess the intracellular traffic of LDL, peritoneal macrophages from 6- to 8-week-old mice were plated onto 8-well µ-slide (ibidi GmbH,

Gewerbehof Gräfelfing, Germany), incubated for 30 min for adherence to the plate, washed twice with PBS, and cultured in RPMI 1640 medium at least 24 h. Cells were then washed with PBS, cultured for 2 h at 37 °C in RPMI 1640 (Lonza Group AG, Basel, Switzerland) without FBS, and incubated for 1 h at 4 °C with 10 μg/mL pHrodo™ Green-LDL (catalog number L34355, Invitrogen™, Thermo Fisher Scientific). Cells were washed four times with 4 °C PBS and cultured at 37 °C for 30 min in RPMI 1640 without FBS (Gibco, Thermo Fisher Scientific) while fluorescence was video-recorded for 30 min in an LSM880 confocal microscope (Zeiss) equipped with 63×/1.4 oil objective. Individual captures and quantification of fluorescence were conducted using ImageJ. To assess oxLDL traffic, macrophages were initially processed as described above for tracking pHrodo™ Green-LDL fluorescence, but after the incubation for 2 h at 37 °C in RPMI 1640 without FBS, cells were incubated for 1 h at 4 °C with 10 μg/mL of DiI-oxLDL (catalog number L34358, Invitrogen™, Thermo Fisher Scientific). Macrophages were then washed four times with 4 °C PBS, and incubated at 37 °C for the indicated time points (10, 30, and 60 min) in RPMI 1640 without FBS. Cells were washed again twice with 4 °C PBS, fixed in 4% PFA, and mounted with ProLong™ Gold Antifade Mountant with DAPI (P36931, Invitrogen™, Thermo Fisher Scientific). Fluorescence was assessed in an LSM880 confocal microscope using the 63×/1.4 oil objective. Individual captures and quantification of DiI fluorescence were conducted using ImageJ.

### Atto-655-oxLDL uptake assay

Peritoneal macrophages were incubated with 50 μg/mL of Atto-655-oxLDL, that was used as tracer for monitoring oxLDL uptake, for 2 h in low attachment plates with RPMI 1640 and then washed twice with PBS. Cells were stained with an Alexa Fluor®488-F4/80 antibody to gate macrophages and the mean fluorescence intensity of Atto-655 was measured in a BD FACSCanto™ II analyzer (BD Biosciences).

### Determination of cytokines by bead-based multiplex array

Serum samples from *Zeb1*^WT^/*Apoe*^KO^ and *Zeb1*^ΔM^/*Apoe*^KO^ mice at the end of the Western diet feeding protocol were used to assess the expression of a panel of 13 cytokines and chemokines (G-CSF, IL-13, IL-2, IL-23p19, IL-4, IL1β, IFNγ, TNFα, MCP-1, IL-6, IL-12p70, IL-17A). Levels of these pro-inflammatory markers were measured by FACS using a bead-based multiplex array (RayPlex™ Mouse Inflammation Array 1, FAM-INF-1-96, RayBiotech Life, Peachtree Corners, GA, USA) as per manufacturer's instructions. Cytokine levels in blood serum were assessed in a BD LSRFortessa™ Cell Analyzer (BD Biosciences), quantified by comparing the fluorescence signal to a standard curve generated from purified protein standards at known concentrations.

### Determination of leptin by ELISA

The leptin concentration was measured in the serum of *Zeb1*^WT^/*Apoe*^KO^ and *Zeb1*^ΔM^/*Apoe*^KO^ mice at the conclusion of the Western diet feeding, utilizing a mouse Leptin ELISA kit (CrystalChem, Elk Grove Village, IL, USA) following the manufacturer's protocol.

### Synthesis, functionalization, and characterization of dendrimer-graphene nanostars

Oxidized graphene nanostars (GNS) (Sigma-Aldrich, reference 804126) were dispersed in DMSO (500 μg/mL) and separated by incubating the dispersion in an ultrasound bath (Selecta, Barcelona, Spain). A hundred μL of GNS were incubated for 2 h with 900 μL of 1 mg/mL EDC/NHS containing 50 μL of PAMAM dendrimer 25% v/v in an ultrasound bath. Afterward, dispersions were centrifuged at 21,000 × g for 10 min, washed one time with DMSO, and four times with PBS. Plasmids (*Lysm*-Ctrl and *Lysm-Zeb1*, see above) were incubated with dispersions of GNS in a ratio of 1:10 for 2 h. Nanoparticle size and Zeta potential were determined by dynamic light scattering (DLS), using a Zetasizer Nano

ZS (Malvern Panalytical, Spectris plc, Malvern, UK). 10 μL of nanoparticle suspension was dispersed in 990 μL of PBS for measurements. DLS data were calculated from the autocorrelation function of scattered light by means of two mathematical approaches: the cumulants method and Dispersion Technology Software nano v. 5.10 (Malvern Panalytical). The mean hydrodynamic diameter (Z-Average) and the width of the particle size distribution (polydispersity index-PDI) were obtained with these approaches, corresponding to approximately 40 measurement runs. Osmolality was determined from osmometric depression of the freezing point in an Advanced™ Micro Osmometer 3300 (Advanced Instruments Inc., Norwood, MA. USA).

### Ex vivo delivery of DGNS into macrophages

Peritoneal macrophages from 6- to 8-week-old mice were isolated and incubated with DGNS-Ctrl or DGNS-*Zeb1* for three days and either left untreated or treated with oxLDL on day two for 24 h. Then, cells were fixed and stained with 1 μg/mL BODIPY® 493/503. Fluorescence was evaluated in a Zeiss Axiovert 200 inverted microscope (Zeiss) and quantified with ImageJ Software. To assess cholesterol efflux, cells were incubated with DGNS-Ctrl or DGNS-Zeb1 and 24 h later cells were treated with 50 μg/mL of oxLDL and 1 μCi/mL [³H]cholesterol for 24 h without changing the medium with the DGNS. Then, cells were equilibrated in serum-free RPMI 1640 medium containing 0.2% (w/v) bovine serum albumin (BSA) for 16 h, followed by two washed in PBS and the incubation with 25 μg/mL lipid-free APOAI (Sigma-Aldrich) or 50 μg/mL HDL for 4 h. After efflux, medium was collected and centrifuged to remove any detached cells. Cells were washed in PBS and lysed in 0.2 M NaOH for 24 h. Medium and cell lysates were counted for radioactivity. The efflux of cholesterol was calculated as per the formula shown above under "Ex vivo cholesterol efflux capacity".

### In vivo delivery of DGNS into mice

Mice subjected to the Western diet protocol were injected every 4 days for a duration of 10 weeks (see schematic in Fig. 6a) with 50 μg of DGNS-Ctrl or DGNS-*Zeb1* (at a 1/10 ratio of plasmid to DGNS) per Kg of body weight. After treatment, mice were euthanized and their aorta and hearts were isolated. Aortas were stained *en face* with 0.5% ORO solution for the quantification of plaque area. For cross-sectional analysis of the plaque, hearts were cut through the aortic sinus in sections of either 7 μm (for H&E staining) or 12 μm (for ORO staining) and three sections of each mouse were stained.

### Statistical analysis

All replicates in this study are biologically independent mouse macrophages, mice, or human samples. Statistical analysis of the data shown in this study was performed using Prism for Mac 8.3 (GraphPad Software, Boston, MA, USA). Bar graphs throughout the manuscript are the mean with standard errors in which the statistical significance was assessed with Mann-Whitney, ANOVA, Student's *t* or Fisher's exact tests. Where appropriate, relevant comparisons were labeled as either significant at the $p \leq 0.0001$ (****), $p \leq 0.001$ (***), $p \leq 0.01$ (**), or $p \leq 0.05$ (*) levels, or non-significant for values of $p > 0.05$, and with specified numerical $p$ values for $0.05 < p < 0.075$. All raw data, along with p values for statistical analyses are included in the Source Data file.

### Reporting summary

Further information on research design is available in the Nature Portfolio Reporting Summary linked to this article.

## Data availability

The RNAseq dataset has been submitted to the Gene Expression Omnibus (GEO) database and assigned accession number GSE206477. Datasets of published RNAseq of human endarterectomies were obtained from GSE41571[45] and GSE163154[68]. Datasets of single cell

RNAseq were obtained from GSE116240[3], GSE149070[45], https://figshare.com/s/c00d88b1b25ef0c5c788 and GSE131778[6] Source data are provided with this paper.

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

## Acknowledgements

We are also indebted to all researchers that generously provided us with reagents (see Supplementary Information). We are grateful to G. Lunazzi (Dr. Hein's group at CRG-CNAG, Barcelona, Spain) for her help in the processing of samples for the RNAseq, and to M. Dabad and B. Martin-Mur (Dr. Esteve-Codina's group at CRG-CNAG) for processing of RNAseq data and assistance in the subsequent bioinformatic analysis, respectively. We thank the Dr. A. García for professional drawing of the schematics in the article (Figs. 1a, 5c, 6a, and Supplementary Fig. S1b, d). We thank the Tumor and Tissue Bank and the Flow Cytometry Unit at IDIBAPS and the Advanced Optical Microscopy and Transmission Electron Microscopy Units at the University of Barcelona School of Medicine and Health Sciences for technical assistance. IDIBAPS is partly funded by the Generalitat of Catalonia's CERCA Programme. The study was conducted at the IDIBAPS' Centre de Recerca Biomèdica Cellex building, which was partly funded by the Cellex Foundation. The different parts of this study were independently funded by grants to A.P. from the Catalan Agency for Management of University and Research Grants (AGAUR) (2021-SGR-01328), and Spanish State Research Agency (AEI) (PID2020-116338RB-IOO) of the Ministry of Science and Innovation (MICINN), as part of the 2021–2023 Plan for Scientific and Technical Research and Innovation (PEICTI), which is co-financed by the European Regional Development Fund (ERDF) of the European Union Commission. A.M.-L. is a recipient of a PhD scholarship from MICINN (FPI Program) (PRE2019-088097). A.E.-C's salary is covered by grant PT17/0009/0019 from Carlos III Spanish Health Institute (ISCIII, co-funded by the ERDF) to CNAG. CNIC is supported by MCINN, ISCIII, the Pro-CNIC Foundation, and is a Severo Ochoa Center of Excellence (CEX2020-001041-S funded by MCIN/AEI/10.13039/501100011033). P.M.-L. salary is supported by a fellowship from AEI's Ramon y Cajal Program (RYC2018-0Z23971-I). L.H. is supported by a PhD scholarship from the China Scholarship Council (202208110041).

## Author contributions

M.C.M.-C performed most of the experimental work in the study; M.C.M.-C and A.P. designed and interpreted experiments; M.C. and C.N. helped with some experiments; A.M.-L. and P.M.-L. synthesized DGNS; L.H. performed scRNAseq analyses on published datasets; V.D. and B.P. generated the *Zeb1*$^{fl/fl}$ (*Zeb1*$^{WT}$) mouse; M.J.A.-M helped in the initial setting of the atherosclerosis mouse model and its analysis; M.F. and J.C.E.-G performed the biochemical analysis of lipids in serum; A.E.-C supervised the RNA-seq processing and the subsequent bioinformatic analysis; FJ.D.-C conducted the pathological analysis of TEM pictures in Fig. 4c; C.E. helped to capture and assessed TEM pictures in Fig. 4i; P.G.-F. and V.A. provided intellectual and technical advice on the project. A.P. conceived, supervised and obtained funding for the study, and wrote the manuscript. All authors provided critical comments on the manuscript.

## Competing interests

The authors declare no competing interests.

## Additional information

$^1$Group of Gene Regulation in Stem Cells, Cell Plasticity, Differentiation, and Cancer, IDIBAPS, 08036 Barcelona, Spain. $^2$Department of Biomedicine, University of Barcelona School of Medicine, 08036 Barcelona, Spain. $^3$Transgenesis Facility, National Center of Biotechnology (CNB) and Center for Molecular Biology Severo Ochoa (UAM-CBMSO), Spanish National Research Council (CSIC) and Autonomous University of Madrid (UAM), Cantoblanco, 28049 Madrid, Spain. $^4$Group of Molecular and Genetic Cardiovascular Pathophysiology, Spanish National Center for Cardiovascular Research (CNIC), 28029

Madrid, Spain. [5]Center for Biomedical, Research Network in Cardiovascular Diseases (CIBERCV), Carlos III Health Institute, 28029 Madrid, Spain. [6]Department of Biochemistry and Molecular Biology, Institute of Biomedical Research Sant Pau, University Autonomous of Barcelona, 08041 Barcelona, Spain. [7]Center for Biomedical Research Network in Diabetes and Associated Metabolic Diseases (CIBERDEM), Carlos III Health Institute, 28029 Madrid, Spain. [8]National Center for Genomics Analysis (CNAG), 08028 Barcelona, Spain. [9]Group of signal transduction, intracellular compartments and cancer, IDIBAPS, 08036 Barcelona, Spain. [10]Department of Pathology, Hospital General Universitario Gregorio Marañón, 28007 Madrid, Spain. [11]Department Of Cell Death and Proliferation, Institute for Biomedical Research of Barcelona (IIBB), Spanish National Research Council (CSIC), 08036 Barcelona, Spain. [12]Group of Hemotherapy and Hemostasis, IDIBAPS, 08036 Barcelona, Spain. [13]Department of Biochemistry and Molecular Genetics, Hospital Clínic, 08036 Barcelona, Spain. [14]Center for Biomedical Research Network in Gastrointestinal and Liver Diseases (CIBEREHD), Carlos III Health Institute, 28029 Madrid, Spain. [15]Institute for Medical Engineering & Science, Massachusetts Institute of Technology (MIT), Cambridge, MA 02139, USA. [16]Molecular Targets Program, Division of Oncology, Department of Medicine, J.G. Brown Cancer Center, Louisville, KY 40202, USA. [17]ICREA, 08010 Barcelona, Spain. [18]These authors contributed equally: Marlies Cortés, Alazne Moreno-Lanceta. ✉e-mail: idib412@recerca.clinic.cat

