## [Peer Review File · Nature Communications]

REVIEWER COMMENTS

Reviewer #1 (Remarks to the Author):

The manuscript described that ZEB1 downregulation reduces cholesterol efflux and augments intracellular lipid accumulation, leading to the formation of larger and more instable atherosclerotic plaques. This paper has demonstrated the mechanistic actions of ZEB1 in macrophages at a molecular and pathological levels, which is of great research value. However, although the work is conceptually interesting and it proposes a potential target for the treatment of atherosclerosis, such a target would have other complications, which was not clearly elaborated. For instance, ZEB1 has been previously studied as a potential target for cancer therapy, and upregulation of ZEB1 may be linked to promoting tumor growth and metastasis. If such a complication can not be properly addressed systemically, this work would not provide meaningful strategy/target for atherosclerotic treatment.

1. In the in vitro experiment of macrophages, the authors only selected two groups of ZEB1 WT and ZEB1 Δ Mac macrophages for validation. Why didn't the authors use RNA interference to silence the expression of target gene? If the authors add this group in the in vitro experiment, the effect of ZEB1 would be further demonstrated. In addition, the effects of ZEB1 on atherosclerotic pathogenesis in wild-type and low-expression mice were studied, but high ZEB1 expression models were not studied, which is actually necessary in order to show the role of ZEB1 on both sides. Meanwhile, ZEB1 is considered as a cancer promoting factor, this is a very important concern and a very major flaw of the work. Sufficient studies and verifications have to be provided.
2. In the result of "ZEB1 inhibits lipid accumulation in macrophages through activation of AMPK 242 signaling", "presence or absence of the AMPK activator A769662" was only BODIPY staining, and it was suggested to increase the mRNA of AMPK targets (Abca1, Abcg1, and Nr1h3, etc.).
3. How stable and biocompatible are GDNS-Ctrl and GDNS- ZEB1? Flow cytometry results in Figure 4A indicate that such nanoparticles may have been taken up by the reticuloendothelial system, most of which are macrophages. Such macrophages are not representative of ZEB1 Δ Mac macrophages. So can GDNS-ZEB1 reach target cells stably?
4. As for the number of data sets in this paper, some figures presented 1 to 2 sets, such as Supplementary Figure 4C, whether such data are representative and reproducible and reliable?
5. There are many formatting and typo errors in the manuscript, for example, there was no unit for the vertical coordinate of Free Cholesterol in FigS1 H; In Line 221, there are two "displayed"; In Line 308, "increases" should be changed to "increased"; In Fig 1L, AUC (area under curve) is suggested to be added.

Reviewer #2 (Remarks to the Author):

The paper of Dr. Postigo on Zeb1 in macrophages and atherosclerosis is interesting. The phenotype is convincing, and the genetic models in combination with the nano therapy model is much appreciated. However, the paper still lacks mechanistic insights. Although many gene expression and in vitro data are reported, insights how Zeb1 affects cholesterol transport, and what is cause vs consequence is unknown. Also, how Zeb1 affects macrophage function, and in which macrophage subtype zeb1 drives macrophage function is not shown.

Specific comments:

Mouse models:

- The analysis of the plaque phenotype lacks crucial information: what is the relative and absolute macrophage content in the plaque? Does macrophage subtype distribution, as reported by single cell analyses (Winkels, Monaco) in the mouse plaque change due to absence of Zeb1?
- Does Zeb1 affect efferocytosis? Since the necrotic core is bigger?
- What is the size of plaque macrophages?
- Is monocyte recruitment affected?
- The WBC data do not give any information on immune cell types or activation, and are not informative. Please perform proper FACS analysis, blood, spleen, (aorta)
- The effects of ZEB1 on glucose tolerance, the effects on liver steatosis, liver weight and the effects on lipid levels suggest a strong metabolic phenotype, please dissect the mechanism more precisely. Insulin resistance? Adipose tissue phenotyping? Is this driven by Kupfer cells or AT macrophages?
- Please show as separate values: total cholesterol, HDL cholesterol, LDL cholesterol, VLDL cholesterol; triglycerides; VLDL in TG fraction. The decrease in HDL also hints towards altered lipid metabolism in the liver, which must be macrophage driven.
- Any signs of lipid accumulation in any of the other organs besides liver and arteries??
- The RNAseq data are informative, but should be supplemented with scRNAseq data to really pinpoint the effects of deficient Zeb1 transcription on macrophage subsets and function.
- Do zeb1 deficient macrophages show defects in phagocytosis, efferocytosis, cytokine production, migration, metabolic handling of nutrients??

Lipid handling:

- A lot of information on how deficiency of Zeb1 in macrophages affects their lipid handling is in the paper, and the data are convincing. However, a precise mechanism on how Zeb1 causes these

disturbances is missing. I would suggest a lipid-loading time course experiment in lipid free medium, including imaging studies. What is first? effects in uptake, endosomal processing? Liposomal processing? Defects in cholesterol sensing?? Or only a defect in cholesterol efflux? The data now are from a stage where lipid loading has occurred, and cause vs consequence are difficult to dissect.

Human data:

-The human data re interesting, but incomplete. In which human plaque macrophage subsets is Zeb1 expressed? How is this correlated with macrophage function? Is it present in the TREM1+ lipid rich macrophages? Or lipid rich IFN+ macrophages? In resident macrophages??

Reviewer #3 (Remarks to the Author):

The manuscript from MC Martinez-Campanario et al., investigates the role of the transcription factor ZEB1 in macrophages in atherosclerotic plaque formation. To examine this, the authors have crossed LysM-Cre mice with ZEB1^{fl/fl} animals and subsequently backcrossed these and CRE- controls onto the APOE-KO background. They demonstrate that loss of Zeb1 from LysM expressing cells leads to increased plaque formation and suggest that this is due to the effect of ZEB1 loss in aortic macrophages. Further experiments suggest that this is regulated by impaired lipid processing in the ZEB1-deficient cells. Overexpression of ZEB1 through nanoparticle targeting reversed this effect, while ZEB1 expression levels also correlate with cardiovascular outcomes in patients. Together this is an interesting study, describing a previously unappreciated role for ZEB1, for which I congratulate the authors. However, in my opinion some conclusions are not yet fully supported by the data provided.

LysM-Cre transgenic mice are used to target myeloid cells, however throughout the text the authors claim these effects are mediated by macrophages. Can the authors confirm there is no role for the other myeloid cells targeted with this CRE line e.g. neutrophils, monocytes, some DCs? If not, then I would strongly suggest the authors tone down their claims that this is mediated by macrophages. Rather I would change the text to say myeloid cells.

For the lipid accumulation & processing studies (Fig 2) the authors use peritoneal macrophages to make their claims. However, exactly how relevant these populations are to the phenotype observed in the plaques is unclear. Can the same findings be replicated in aortic macrophages?

In figure 3A, the authors perform RNA-seq (bulk) analysis on aortic macrophages. However, as this is bulk analysis, this raises the question of what is being compared. Does loss of ZEB1 affect the different populations of macrophages present in the aorta? Can the authors be sure they are comparing the same cells in WT and KO? Or does the KO affect recruitment of monocyte-derived macrophages or perhaps

regulate the presence of resident macrophages. Given the proposed different contributions of the various populations to the development of plaques, any heterogeneity would be important to clarify.

In figure 1 the authors show that these mice also have larger livers with more lipid deposits than WT controls. Is this also driven by macrophages? What happens to the different subsets of hepatic macrophages in these mice? Moreover, in Figure 5, the authors demonstrate the ZEB1 overexpression reverses the plaque phenotype but does this also reverse the systemic weight gain and the effects on the liver? How much of this phenotype can be attributed to aortic macrophages versus systemic effects? On that front, can the authors also demonstrate that the mice consume similar quantities of food and have similar metabolic outputs as controls?

Minor comments

Fig 1C appears to be incorrectly annotated as Zeb1-macs rather than Zeb1+/-?

Sirius red should also be quantified in Fig 1I

Line 205 (of for Bodipy) typo

Has the effect of the CRE been determined? E.g. with CRE only controls? This is especially important for systemic effects where the effects of ZEB1 overexpression were not discussed.

RESPONSE TO REVIEWERS

A) REPLY TO REVIEWER #1

The manuscript described that ZEB1 downregulation reduces cholesterol efflux and augments intracellular lipid accumulation, leading to the formation of larger and more instable atherosclerotic plaques. This paper has demonstrated the mechanistic actions of ZEB1 in macrophages at a molecular and pathological levels, which is of great research value. However, although the work is conceptually interesting and it proposes a potential target for the treatment of atherosclerosis, such a target would have other complications, which was not clearly elaborated. For instance, ZEB1 has been previously studied as a potential target for cancer therapy, and upregulation of ZEB1 may be linked to promoting tumor growth and metastasis. If such a complication can not be properly addressed systemically, this work would not provide meaningful strategy/target for atherosclerotic treatment.

We greatly appreciate Reviewer #1's positive comments regarding the research value and interest of the study.

The Reviewer's concern about how the tumor-promoting role of ZEB1 may impact its potential use as a target in the treatment of atherosclerosis is addressed in our response to point 1 below.

1) In the *in vitro* experiment of macrophages, the authors only selected two groups of ZEB1 WT and ZEB1^{ΔMac} macrophages for validation. Why didn't the authors use RNA interference to silence the expression of target gene? If the authors add this group in the *in vitro* experiment, the effect of ZEB1 would be further demonstrated. In addition, the effects of ZEB1 on atherosclerotic pathogenesis in wild-type and low-expression mice were studied, but high ZEB1 expression models were not studied, which is actually necessary in order to show the role of ZEB1 on both sides. Meanwhile, ZEB1 is considered as a cancer promoting factor, this is a very important concern and a very major flaw of the work. Sufficient studies and verifications have to be provided.

Regarding the use of RNA interference to silence gene expression in *in vitro* experiments.

- We may not have completely understood Reviewer #1's comment and the specific figure he/she is referring to. We have provided our response to the best of our understanding, but we apologize in advance if it does not align with Reviewer #1's request. We are open to running any additional experiments Reviewer #1 may suggest if we did not fully understand what she/he requested or if further clarification is needed.
- Based on Reviewer #1's comment regarding the use of RNA interference to silence target genes in the *in vitro* experiments, we interpreted that he/she suggested eliminating AMPK in the original Figure 2 (now Figure 3). As primary macrophages are not easily transfected with siRNAs, we employed a pharmacological inhibitor of AMPK, compound C, to silence AMPK target genes in macrophages. Compound C increased lipid accumulation in *Zeb1*^{WT} macrophages, as assessed by BODIPY, to similar levels as in *Zeb1*^{ΔM} macrophages (Figure 3H).

➔ EXPERIMENTS CONDUCTED: Following Reviewer #1's comment to silence target genes in *in vitro* experiments, we use compound CC, a pharmacological inhibitor of AMPK (Figure 3H).

Regarding the role of ZEB1 as a tumor-promoting factor. We are grateful to Reviewer #1 for the possibility to clarify this important issue.

- ZEB1 is not an oncogene in epithelial cells. Work by our group and others showed that ZEB1 can reprogram a cancer cell toward a stem-like, invasive and chemo-resistant phenotype (*Nat Commun* 9:2424; *Nat Commun*. 5:5660, reviewed in *EMBO J*. 40:e108647). However, by itself, ZEB1 does not have an oncogenic effect on a normal cell in the absence of underlying oncogenic mutations (RAS, APC/Wnt, etc.). As a mesenchymal marker and EMT driver, ZEB1 is not expressed by epithelial cells, which are the origin cells of carcinomas. In summary, ZEB1 expression alone is insufficient to convert a normal epithelial cell into a malignant one and form a carcinoma.
- ZEB1 is expressed and essentially required in the physiological/homeostatic functions of non-epithelial cells. ZEB1 is not only expressed by non-epithelial cells of multiple origins (from immune cells to muscle cells), but ZEB1 downregulation in these cells is linked to

multiple physiopathological and disease conditions (e.g., muscle atrophy, muscular dystrophy, osteoporosis, Alzheimer, corneal dystrophy, and in this study, plaque formation) (e.g., *Nat Commun*, 10:1364; *Nucleic Acids Res.* 46:10697-708; *Cell Genomics* 3:100263; *Ann. Hum. Genet.*, 79:1-9; *Nat Commun.* 11:460).

- Stably and transient overexpression of ZEB1 in therapy. As evidence of the role of ZEB1 expression in non-epithelial cells has begun to emerge, several studies have overexpressed ZEB1 in vivo, stably (using transgenic models) and transiently. For instance, in *Nature Communications* 11:460, Wu and colleagues administered ZEB1-packaged liposomes to treat osteoporosis. Importantly, our approach to upregulate ZEB1 differs from these previous strategies because we specifically targeted the expression of ZEB1 to our cell of interest. We used macrophage-targeted nanoparticles containing the *LysM* promoter as the driver of the *Zeb1* cDNA in these nanoparticles (referred to in the study as GDNS-*Zeb1*) and compared them with nanoparticles with the *LysM* promoter driving a non-encoding scrambling sequence (referred to in the study as GDNS-Ctrl). This means that ZEB is expressed exclusively in differentiated myeloid cells, and not in epithelial cells, or any non-myeloid cell.
- Effect of macrophage-targeted GDNS-*Zeb1* nanoparticles in other tissues. The arguments above would rule out that ZEB1 by itself can generate a malignant cell. Formal proof that our GDNS-*Zeb1* nanoparticles do not induce tumor formation would require treating mice with *LysM* control and *LysM-Zeb1* nanoparticles for 12-18 months and then examining potential tumor development using PET scans. After discussing this with the Editor and considering the safeguards regarding the specificity of the GDNS-*Zeb1* nanoparticles, we concur that delaying the paper by 12-18 months is not warranted. Nevertheless, in response to the comments from this Reviewer, we conducted additional experiments to show that GDNS-*Zeb1* nanoparticles do not induce ZEB1 expression in cells outside the myeloid lineage. We treated wild-type mice with GDNS-Ctrl and GDNS-*Zeb1* nanoparticles, euthanized and selected organs (pancreas, colon, lung, and liver) were analyzed for ZEB1 expression by immunofluorescence (Figure A). We found that ZEB1 was not expressed in any of the epithelial cells in these organs. Please, note that staining in the lumen of the colonic crypts is non-specific and corresponds to the capture of the antibody by the lumen mucus.
- Given that these results have no bearing on the data shown in the manuscript, we considered that it would be confusing to integrate it into the main manuscript and have included Figure 1 in this letter but not in the manuscript. However, if Reviewer #1 considers it appropriate, it can be included in the Supplementary Information.

Figure A: representative images of ZEB1 immunofluorescence staining in the pancreas, colon, lung, and liver from wild-type mice treated with GDNS-Ctrl and GDNS-Zeb1 nanoparticles.

2) In the result of "ZEB1 inhibits lipid accumulation in macrophages through activation of AMPK 242 signaling", "presence or absence of the AMPK activator A769662" was only BODIPY staining, and it was suggested to increase the mRNA of AMPK targets (*Abca1*, *Abcg1*, and *Nr1h3*, etc.).

We thank Reviewer #1 for her/his constructive suggestion.

Following the comments by this Reviewer, we analyzed the effect of the AMPK activator A769662 on the expression of AMPK targets *Ppargc1a* and *Nr1h3*. These data are now shown in the new Figure 3G.

→ Following the comments by Reviewer #1, the revised manuscript examined the effect of the A769662 on the expression of AMPK targets (Figure 3G).

3) How stable and biocompatible are GDNS-Ctrl and GDNS- ZEB1? Flow cytometry results in Figure 4A indicate that such nanoparticles may have been taken up by the reticuloendothelial system, most of which are macrophages. Such macrophages are not representative of ZEB1 Δ Mac macrophages. So can GDNS-ZEB1 reach target cells stably?

We thank Reviewer #1 for her/his comments and the possibility to clarify this important point.

- GDNS-Ctrl and GDNS-Zeb1 nanoparticles are graphene nanostars, specifically single-wall carbon nanohorns, which have been used for drug and gene delivery by co-authors in this study and other researchers (*Sci Transl Med.* 15:eabq62; *Nat Commun*, 9:2393; *Nano Lett*, 18:5839-5845, *Nanotechnology* 22:265106; *ACS Nano.* 2:213-26). These studies demonstrate that graphene nanostars exhibit higher biocompatibility with macrophages and other cell types compared to other nanoparticles (*Nat Commun*, 9:2393; patent application 18163253). In addition, the long-term intravenous administration of graphene nanostars in mice, up to 26 weeks, showed no signs of toxicity (*ACS Nano* 2:213-226; *Nanotechnology* 22:265106). In the revised manuscript, we have added a note to discuss this issue (pages 14-15)
- Human monocytes in the peripheral blood have an average lifespan of 1-7 days (*J Exp Med*, 214:1913-1923), while macrophages within the atherosclerotic plaque have a turnover rate of less than 4 weeks (*Nat Medicine*, 19:1166-72). As a result, any modulation of Zeb1 using GDNS-Zeb1 nanoparticles (or any other delivery method developed in the future) would necessitate their regular administration to maintain the effects as shown by other articles using nanoparticles for the treatment of atherosclerosis and other pathologies (e.g., *Nat Commun* 5:3065; *Nat Nanotech* 15:154-61; *ACS Nano* 12:8943-60; *Sci Transl Med*, 15:eabq6225).
- While we cannot rule out that other phagocytes of the reticuloendothelial system also phagocyte the GDNS nanoparticles, we found that virtually all macrophages phagocytosed DGNS nanoparticles, while almost none of the non-macrophage cells did it (Figure 5A). We also found that intravenous injection of DGNS efficiently accumulated in the aorta (Figure 5B).
- In addition, we have now examined the localization of macrophages in the aorta of Zeb1 Δ M treated with GDNS-Ctrl and GDNS-Zeb1 nanoparticles. Our results indicated that ZEB1 expression was targeted to macrophages, as assessed by the expression of LAMP2 (MAC-3), within the atherosclerotic plaque (new Figure 6C).

➔ Following the comments by Reviewer #1, the revised manuscript has implemented the following changes: a) the biocompatibility and lack of toxicity *in vivo* of graphene nanostars are discussed (pages 14-15); and b) we examined the localization of macrophages in the aorta of Zeb1 Δ M mice following treatment with GDNS-Ctrl and GDNS-Zeb1 nanoparticles (Figure 6C).

4) As for the number of data sets in this paper, some figures presented 1 to 2 sets, such as Supplementary Figure 4C, whether such data are representative and reproducible and reliable?

We thank Reviewer #1 for her/his comment on the original Supplementary Figure S4C (now Supplementary Figure S5B). We have performed additional experiments to increase the number of replicates in the new Supplementary Figure S5B as well as in other figures in the study, namely Figures 5D, 5E, 6D, 6E, and 6F .

→ Following the comments by Reviewer #1, in the revised manuscript we have increased the number of replicates in the original Supplementary Figure S4C (now Supplementary Figure S5B) and other figures of the study (5D, 5E, 6D, 6E, and 6F).

5) There are many formatting and typo errors in the manuscript, for example, there was no unit for the vertical coordinate of Free Cholesterol in FigS1 H; In Line 221, there are two "displayed"; In Line 308, "increases" should be changed to "increased"; In Fig 1L, AUC (area under curve) is suggested to be added.

We thank Reviewer #1 for pointing out these formatting and typographical errors. The manuscript has been revised throughout and these issues have been corrected in the revised manuscript. The AUC for the former Figure 1L (now Figure 2O) is shown in the Supplementary Figure S2E.

C) REPLY TO REVIEWER #2

The paper of Dr. Postigo on Zeb1 in macrophages and atherosclerosis is interesting. The phenotype is convincing, and the genetic models in combination with the nano therapy model is much appreciated. However, the paper still lacks mechanistic insights. Although many gene expression and in vitro data are reported, insights how Zeb1 affects cholesterol transport, and what is cause vs consequence is unknown. Also, how Zeb1 affects macrophage function, and in which macrophage subtype zeb1 drives macrophage function is not shown.

We sincerely appreciate Reviewer #2's positive feedback regarding the interests and technological approaches employed in our study.

In response to the Reviewer's inquiry regarding the mechanisms by which ZEB1 regulates macrophage functions, including cholesterol transport, we have addressed them in the following sections.

Specific comments:

1) Mouse models:

1.1) The analysis of the plaque phenotype lacks crucial information: what is the relative and absolute macrophage content in the plaque? Does macrophage subtype distribution, as reported by single cell analyses (Winkels, Monaco) in the mouse plaque change due to absence of Zeb1?

We thank Reviewer #2 for her/his constructive comments.

- In the revised manuscript we have quantified the absolute number of macrophages (area of macrophages/field) and their proportion relative to other CD45+ immune cells. We found that the atherosclerotic plaques of *Zeb1*^{ΔM}/*Apoe*^{KO} mice contained more macrophages (assessed by MAC3 expression) compared to the plaques of *Zeb1*^{WT}/*Apoe*^{KO} mice (Figure 1M). Moreover, the proportion of macrophages among all immune cells was also higher in the atherosclerotic plaques of *Zeb1*^{ΔM}/*Apoe*^{KO} mice compared to the plaques of *Zeb1*^{WT}/*Apoe*^{KO} mice (Figure 2B). As anticipated, macrophages in the atherosclerotic plaques of *Zeb1*^{ΔM}/*Apoe*^{KO} did not express ZEB1, while those in *Zeb1*^{WT}/*Apoe*^{KO} plaques did express ZEB1 (Supplementary Figure S1F).
- We have also examined ZEB1 expression in publicly available scRNAseq data of mouse and human atherosclerosis (*Cardiovasc Res*, 119:1676-1689). Our analysis revealed that ZEB1 is differentially expressed among various macrophage subpopulations within the mouse and human atherosclerotic plaque. Specifically, it showed higher expression levels of Zeb1 in inflammatory macrophages compared to other subpopulations (Figure 2A and Supplementary Figure S2C).
- Naturally, none of the published scRNAseq studies of mouse atherosclerosis have explored whether the absence of Zeb1 affects the distribution of macrophage subtypes in the plaque. While it would have been interesting to conduct, analyze, and validate scRNAseq of *Zeb1*^{WT}/*Apoe*^{KO} and *Zeb1*^{ΔM}/*Apoe*^{KO} mice and examine the distribution of macrophage subpopulations, this would require much more time than the journal's deadline allows.
- As explained in greater detail in our response to point 1.9), we have conducted FACS analyses to determine the proportions of total macrophages (CD45⁺, CD11b⁺, F4/80⁺) as well as CD86⁺ and CD9⁺ macrophages within the plaques of *Zeb1*^{WT}/*Apoe*^{KO} and *Zeb1*^{ΔM}/*Apoe*^{KO} mice (Figures 2B-2H). See also below in point 1.9.

➔ EXPERIMENTS CONDUCTED: Following the comments by Reviewer #2, we have implemented the following changes: a) we have quantified the absolute and relative number of macrophages in the plaque of *Zeb1*^{WT}/*Apoe*^{KO} and *Zeb1*^{ΔM}/*Apoe*^{KO} mice (Figures 1J and 2B); and b) analyzed published scRNAseq datasets to explore the differential expression of ZEB1 among

macrophage subpopulations in the plaque of mouse and human models of atherosclerosis in published scRNAseq datasets (Figure 2A and Supplementary Figure S2C); and c) examined by FACS the share of aortic CD86⁺ inflammatory and CD9⁺ foam macrophage in *Zeb1*^{WT}/*ApoE*^{KO} and *Zeb1*^{ΔM}/*ApoE*^{KO} mice (Figures 2B-2H).

1.2) Does *Zeb1* affect efferocytosis? Since the necrotic core is bigger?

We thank Reviewer #2 for her/his constructive comments to improve the study.

- We have quantified the necrotic core in the plaques of *Zeb1*^{WT}/*ApoE*^{KO} and *Zeb1*^{ΔM}/*ApoE*^{KO} mice. As shown in Figure 1L, *Zeb1*^{ΔM}/*ApoE*^{KO} plaques have a larger necrotic core than *Zeb1*^{WT}/*ApoE*^{KO} counterparts.
- We also quantified efferocytosis in the plaques of both genotypes. Compared with *Zeb1*^{WT}/*ApoE*^{KO} counterparts, *Zeb1*^{ΔM}/*ApoE*^{KO} plaques exhibited a deficient efferocytosis as assessed by the quantification of the “macrophage-associated apoptotic cells” to “free apoptotic cells” (Figure 1M).

→ Following the comments by Reviewer #2, the revised manuscript has implemented the following changes: a) we have quantified the necrotic core in *Zeb1*^{WT}/*ApoE*^{KO} and *Zeb1*^{ΔM}/*ApoE*^{KO} plaques (Figure 1L); and b) we have quantified efferocytosis in the plaques of both genotypes (Figure 1M).

1.3) What is the size of plaque macrophages?

We also thank Reviewer #2 for raising this point.

We have found that *Zeb1*^{ΔM}/*ApoE*^{KO} plaques contain more macrophages compared to *Zeb1*^{WT}/*ApoE*^{KO} counterparts. Unfortunately, we encountered difficulties in devising a reliable quantification method to assess the size of intra-plaque *Zeb1*^{WT} and *Zeb1*^{ΔM} macrophages due to macrophages clustering together. We are open to suggestions by Reviewer #2 for conducting any additional experiments to address this point if considered important for the publication of the study.

1.4) Is monocyte recruitment affected?

The higher number of macrophages in *Zeb1*^{ΔM}/*ApoE*^{KO} plaques compared to *Zeb1*^{WT}/*ApoE*^{KO} plaques (Figure 1J), which could be occurring simultaneously and that, within the existing time limitations, we have tried to explore.

As detailed below, we conducted an analysis of inflammatory markers in the serum *Zeb1^{WT}/ApoE^{KO}* and *Zeb1^{ΔM}/ApoE^{KO}* mice and found that the latter expressed higher levels of the chemokine CCL2, which is a pathogenic factor in the recruitment of macrophages into the atherosclerotic plaque and has been associated with plaque vulnerability and cardiovascular accidents (*Circulation* 139:256-268; *Arterioscler Thromb Vasc Biol*, 41:2038-2048). Following that result, we decided to examine monocyte migration into the atherosclerotic plaque using fluorescence latex beads (Fluoresbrite® YG Microspheres 1.00μm, Polyscience, Warrington, PA). We injected the beads into *Zeb1^{WT}/ApoE^{KO}* and *Zeb1^{ΔM}/ApoE^{KO}* mice, extracted the aortas and processed them by IDISCO protocol (*Cell*, 159:896-910). Regrettably, the experiments did not work technically as specific bead fluorescence was not observed in the plaques of mice from either genotype. We do not understand the reason why this technique did not work in our hands.

1.5) The WBC data do not give any information on immune cell types or activation, and are not informative. Please perform proper FACS analysis, blood, spleen, (aorta)

We thank Reviewer #2 for her/his constructive suggestion.

Following Reviewer #2's comments we analyzed by FACS the distribution of myeloid cells in the spleen. The reason behind this choice of tissue is that peripheral blood and aorta contain fewer cells to conduct a comprehensive FACS analysis. As noted above, we have now examined the distribution of myeloid cells within the CD11b⁺ cells of the spleen. We found that *Zeb1^{WT}/ApoE^{KO}* and *Zeb1^{ΔM}/ApoE^{KO}* mice exhibit similar shares of CD11c⁺, F4/80⁺, Gr1⁺, and Ly6C⁺ cells (Supplementary Figure S2B).

→ Following the comments of Reviewer #2, we have examined the distribution of myeloid cell populations by FACS in the spleen of *Zeb1^{WT}/ApoE^{KO}* and *Zeb1^{ΔM}/ApoE^{KO}* mice (Supplementary Figure S2B).

1.6) The effects of ZEB1 on glucose tolerance, the effects on liver steatosis, liver weight and the effects on lipid levels suggest a strong metabolic phenotype, please dissect the mechanism more precisely. Insulin resistance? Adipose tissue phenotyping? Is this driven by Kupfer cells or AT macrophages?

We thank Reviewer #2 for her/his comments.

As this Reviewer indicates, results in the original manuscript suggested that *Zeb1^{ΔM}/ApoE^{KO}* mice have an altered systemic metabolic phenotype. To further define the phenotype and mechanism of action by which ZEB1 regulate it, in the revised manuscript, we have conducted the following experiments.

- The serum levels of a panel of cytokines were examined in *Zeb1^{WT}/ApoE^{KO}* and *Zeb1^{ΔM}/ApoE^{KO}* mice using a commercial multiplexed sandwich- and bead-based quantitative

antibody array commercial kit (RayPlex® Mouse Inflammation Array Kit 1, RayBiotech Life, Inc.), which was assessed by FACS. As shown in the new Figure 2M, the sensitivity of this array was able to detect 4 of the 13 cytokines included in the kit, namely IL1 β , IL2, IL12, and CCL2. *Zeb1* ^{Δ M}/*Apoe*^{KO} mice expressed higher serum levels of these cytokines compared to *Zeb1*^{WT}/*Apoe*^{KO} mice.

- Compared to *Zeb1*^{WT}/*Apoe*^{KO} livers, the livers of *Zeb1* ^{Δ M}/*Apoe*^{KO} mice exhibited higher expression of the nuclear processed form of SREBP1c (Figure 2S and Supplementary Figure S2G). Nuclear SREBP1c triggers the activation of liver enzymes involved in *de novo* lipogenesis, leading to the accumulation of fatty acids in the liver and higher lipid serum levels (reviewed in *Cell*, 184: 2537-2564, *Biochem J.* 478:3723-39).
- We also examined the potential role of ZEB1 in white adipose tissue (WAT)-mediated inflammation. WAT is a major source of systemic inflammation through the activities of adipocytes themselves as well as of infiltrating immune cells, chiefly macrophages (reviewed in *Circ Res.* 128:951-68). Compared to *Zeb1*^{WT}/*Apoe*^{KO} mice, the WAT of *Zeb1* ^{Δ M}/*Apoe*^{KO} mice exhibited higher infiltration of macrophages (Figure 2L). In *Zeb1* ^{Δ M}/*Apoe*^{KO} mice, infiltrating macrophages were effectively knock out for *Zeb1* (Figure 2J) and expressed higher *Il1b* mRNA levels compared to WAT macrophages in *Zeb1*^{WT}/*Apoe*^{KO} (Figure 2K).
- We also found that serum levels of leptin were higher in *Zeb1* ^{Δ M}/*Apoe*^{KO} mice than in *Zeb1*^{WT}/*Apoe*^{KO} mice (Figure 2N). In addition to myocardium and perivascular adipose tissue, WAT (mainly its adipocytes) is a major source of the adipokine leptin (*Atherosclerosis* 189:47-60; *Nat Rev Cardiol.* 7:22-29). While leptin exhibits multifaceted effects on atherosclerosis in both humans and mice, leptin promotes the recruitment of monocytes into the arterial intima, the formation of foam cells, the proliferation of vascular smooth muscle cells, and the secretion of classical pro-inflammatory cytokines by adipocytes and macrophages.

➔ Following the comments by Reviewer #2, the revised manuscript has implemented the following changes: a) we have quantified serum levels of cytokines in *Zeb1*^{WT}/*Apoe*^{KO} and *Zeb1* ^{Δ M}/*Apoe*^{KO} mice (Figure 2M); b) analyzed SREBP1c in the livers of both genotypes (Figure 2S and Supplementary Figure S2G), and c) examined by infiltration by macrophages of the white adipose tissue in *Zeb1*^{WT}/*Apoe*^{KO} and *Zeb1* ^{Δ M}/*Apoe*^{KO} mice as well as macrophage production of *Ilb* (Figure 2K and 2L); and d) assessed serum levels of leptin in *Zeb1* ^{Δ M}/*Apoe*^{KO} and *Zeb1*^{WT}/*Apoe*^{KO} mice (Figure 2N).

1.7) Please show as separate values: total cholesterol, HDL cholesterol, LDL cholesterol, VLDL cholesterol; triglycerides; VLDL in TG fraction. The decrease in HDL also hints towards altered lipid metabolism in the liver, which must be macrophage driven.

We thank Reviewer #1 for the chance to clarify this point.

- Separate values of total cholesterol, HDL cholesterol, and triglycerides. Separate values of these parameters in the serum of *Zeb1*^{WT}/*ApoE*^{KO} and *Zeb1*^{ΔM}/*ApoE*^{KO} mice were presented in the former Figure 1K and Supplementary Figure S1H, which are now Figure 2T and Supplementary Figure S2H in the revised manuscript.
- Separate values of LDL. We have now determined the values for LDL in the serum of *Zeb1*^{WT}/*ApoE*^{KO} and *Zeb1*^{ΔM}/*ApoE*^{KO} mice (Figure 2T).

→ Following the comments by Reviewer #2, we have determined the values for LDL in the serum of *Zeb1*^{WT}/*ApoE*^{KO} and *Zeb1*^{ΔM}/*ApoE*^{KO} mice (Figure 2T).

1.8) Any signs of lipid accumulation in any of the other organs besides liver and arteries??

Following the suggestion by Reviewer #2, we have examined signs of lipid accumulation in skeletal muscles and kidney.

- Staining with Oil Red O in the skeletal muscle of *Zeb1*^{WT}/*ApoE*^{KO} and *Zeb1*^{ΔM}/*ApoE*^{KO} mice indicates higher lipid deposits in the latter (Supplementary Figure S2I).
- Staining with Oil Red O in the kidneys of *Zeb1*^{WT}/*ApoE*^{KO} and *Zeb1*^{ΔM}/*ApoE*^{KO} mice indicates a small increase, although not statistically significant, of lipid deposits in the latter (Supplementary Figure S2J).

→ Following the comments by Reviewer #2, the revised manuscript has examined the accumulation of lipids in the skeletal muscles and kidneys of *Zeb1*^{WT}/*ApoE*^{KO} and *Zeb1*^{ΔM}/*ApoE*^{KO} mice (Supplementary Figure S2I and 2J).

1.9) The RNAseq data are informative, but should be supplemented with scRNAseq data to really pinpoint the effects of deficient Zeb1 transcription on macrophage subsets and function.

We thank Reviewer #2 for her/his constructive comment.

- We fully concur with the Reviewer that conducting, analyzing, and validating a scRNAseq of aortic plaque cells from *Zeb1*^{WT}/*ApoE*^{KO} and *Zeb1*^{ΔM}/*ApoE*^{KO} mice would have been of great interest. However, not only do these types of experiments take up to a year.

- We have explored ZEB1 expression in publicly available scRNAseq data from mouse and human atherosclerosis studies (*Cardiovasc Res*, 119:1676-1689; *Nat Med.* 25:1576-1588; *Nat Med.* 25:1280-1289. *Circ Res.* 123:1127-1142; *Circ Res.* 122:1675-1688). Obviously, none of the scRNAseq studies on mouse atherosclerosis have the possibility to study the effect of *Zeb1* ablation on the distribution of macrophage subtypes within the plaque. Nonetheless, our analysis supports the data shown in this manuscript. ZEB1 expression patterns varies among various macrophage subpopulations within both mouse and human atherosclerotic plaques. Specifically, we observed elevated expression levels of *Zeb1* in inflammatory macrophages compared to other subpopulations (Figure 2A and Supplementary Figure S2C).
- In the original manuscript, we showed that *Zeb1* ablation did not influence the overall distribution of peripheral blood immune populations (original Supplementary Figure S1F, now Supplementary Figure S2A).
- We found that the macrophages represented a higher proportion out of CD45⁺ immune cells within the *Zeb1*^{ΔM}/*ApoE*^{KO} plaques than within *Zeb1*^{WT}/*ApoE*^{KO} plaques (Figure 2B). The share of CD86⁺ macrophages out of total intra-plaque macrophages was similar in both genotypes (Figures 2C and 2E). However, in line with the higher number of macrophages infiltrating the plaques of *Zeb1*^{ΔM}/*ApoE*^{KO} mice, the share of CD86⁺ macrophages out of total CD45⁺ immune cells infiltrating the plaques of *Zeb1*^{ΔM}/*ApoE*^{KO} mice was higher than in the plaques of *Zeb1*^{WT}/*ApoE*^{KO} mice (Figures 2D and 2E). In other words, compared to *Zeb1*^{WT}/*ApoE*^{KO} plaques, *Zeb1*^{ΔM}/*ApoE*^{KO} plaques contain a higher proportion of total macrophages and inflammatory macrophages out of all CD45⁺ immune cells.
- In line with the higher number of lipid-loaden macrophages in *Zeb1*^{ΔM}/*ApoE*^{KO} plaques (Figures 1I and 4C) the share of CD9⁺ foam macrophages out of total intra-plaque macrophages was higher in *Zeb1*^{ΔM}/*ApoE*^{KO} plaques in comparison to *Zeb1*^{WT}/*ApoE*^{KO} plaques (Figures 2F-2H).

→ Following the comments by Reviewer #2, the revised manuscript has implemented the following changes: a) we examined ZEB1 expression in publicly available scRNAseq data from mouse and human atherosclerosis studies (Figure 2A and Supplementary Figure S2C); and b) we analyzed by FACS the proportion of inflammatory (CD86⁺) and foam (CD9⁺) macrophages within the atherosclerotic plaques of *Zeb1*^{WT}/*ApoE*^{KO} and *Zeb1*^{ΔM}/*ApoE*^{KO} mice (Figures 2B-2H).

1.10) Do zeb1 deficient macrophages show defects in phagocytosis, efferocytosis, cytokine production, migration, metabolic handling of nutrients??

We thank the Reviewer for her/his suggestion to analyze these five processes. We recognize that exploring them can potentially provide important mechanistic insights into how ZEB1 expression in macrophages explains the phenotype of *Zeb1*^{ΔM}/*ApoE*^{KO} mice. However, addressing each one of them can be an article or several articles on their own.

Within the existing time restrictions, we have conducted the following experiments to strengthen the study. Of course, we are open to conducting any additional experiments that Reviewer #2 may propose to explore these processes in a straightforward manner if she/he considers it is important for the publication of the manuscript.

- Efferocytosis. As shown in Figure 1M, *Zeb1*^{ΔM}/*ApoE*^{KO} plaques have more free apoptotic cells than apoptotic cells associated to macrophages compared with *Zeb1*^{WT}/*ApoE*^{KO} counterparts.
- Cytokine production. As noted in point 1.6), we examined the serum levels of a cytokine/chemokine panel in both *Zeb1*^{WT}/*ApoE*^{KO} and *Zeb1*^{ΔM}/*ApoE*^{KO} mice using a commercial multiplexed sandwich- and bead-based quantitative antibody array kit (Figure 2M)
- Migration. As noted in point 1.4), we found higher levels of the chemokine CCL2 in the serum of *Zeb1*^{ΔM}/*ApoE*^{KO} mice compared to the serum of *Zeb1*^{WT}/*ApoE*^{KO} mice (Figure 2M).

→ Following the comments by Reviewer #2, the revised manuscript has implemented the following changes: a) we have determined the presence of efferocytosis in the plaques of *Zeb1*^{WT}/*ApoE*^{KO} and of *Zeb1*^{ΔM}/*ApoE*^{KO} mice (Figure 1M); and b) we determined the expression of a panel of inflammatory cytokines/chemokines, including the chemokine CCL2, in the serum of *Zeb1*^{WT}/*ApoE*^{KO} and *Zeb1*^{ΔM}/*ApoE*^{KO} mice (Figure 2M).

2) Lipid handling:

A lot of information on how deficiency of Zeb1 in macrophages affects their lipid handling is in the paper, and the data are convincing. However, a precise mechanism on how Zeb1 causes these disturbances is missing. I would suggest a lipid-loading time course experiment in lipid free medium, including imaging studies. What is first? effects in uptake, endosomal processing? Liposomal processing? Defects in cholesterol sensing?? Or only a defect in cholesterol efflux? The data now are from a stage where lipid loading has occurred, and cause vs consequence are difficult to dissect.

We thank Reviewer #2 for the chance to clarify this important point.

- We would like to note that the time course of lipid transport in *Zeb1*^{WT} and *Zeb1*^{ΔM} macrophages culture in lipid free medium requested by the Reviewer was previously presented in the original Figure 3D-3G (now Figure 4D-4G). We apologize if the data presentation was not clear enough. In the revised manuscript, we have rewritten the corresponding paragraph and figure legends (pages 13 and 14). Additionally, we have included two videos showing the *in vivo* time course of phrodo-LDL intracellular transport in *Zeb1*^{WT} and *Zeb1*^{ΔM} macrophages in lipid-free medium (videos uploaded as Supplementary Materials).
- In the revised manuscript, we examined macrophages from both genotypes by transmission electron microscopy (TEM) after internalizing oxLDL (Fig. 4I). *Zeb1*^{WT} macrophages exhibited prototypical multivesicular bodies (late endosomes), indicative of normal uptake and transport of oxLDL. Instead, consistent with the data in Fig. 4D-G, *Zeb1*^{ΔM} macrophages aberrantly accumulate cholesterol in the late endocytic compartment (lysosomes), some in the form of cholesterol crystals (Fig. 4I), which are known to trigger the NLRP3 inflammasome (*Nature*. 464:1357-61; *Circ Res*. 122:1722-40). *Zeb1*^{ΔM} macrophages also showed signs of Golgi fragmentation (Fig. 4I), which supports an intracellular trafficking malfunction, both in the exocytic pathway and the biogenesis of lysosomes.
- The data shown in the original Figures 2 and 3 (now Figures 3 and 4) suggest that *Zeb1*^{ΔM}/*Apoe*^{KO} macrophages undergo alterations in the intracellular transport of lipids affecting cholesterol along the endocytic and exocytic routes that are involved in atherosclerotic plaque formation. Two DEGs in the RNAseq related with vesicular trafficking were downregulated in *Zeb1*^{ΔM}/*Apoe*^{KO} macrophages, namely *Vps52* (a member of the GARP complex) and *Chmp1b* (ESCRT-III complex) (Figure 4B).
- From our data, we concluded that endocytosis is not compromised in *Zeb1*^{ΔM}/*Apoe*^{KO} macrophages as evidenced by their uptake of LDL and oxLDL in those macrophages during the first 10 min (Figures 4D-4G).

→ Following the comments by Reviewer #2, the revised manuscript has implemented the following changes: a) we have now included video files showing the *in vivo* time course of phrodo-LDL intracellular transport in *Zeb1*^{WT} and *Zeb1*^{ΔM} macrophages in lipid-free medium (videos uploaded as Supplementary Materials); and b) we examined macrophages from both genotypes by transmission electron microscopy (TEM) after internalizing oxLDL to examine the formation of late endosomes and signs of lysosomal damage (Fig. 4I).

3) Human data:

The human data re interesting, but incomplete. In which human plaque macrophage subsets is Zeb1 expressed? How is this correlated with macrophage function? Is it present in the TREM1+ lipid rich macrophages? Or lipid rich IFN+ macrophages? In resident macrophages??

We thank this Reviewer for the chance to discuss this relevant issue.

- As noted in point 1.9), we have also analyzed three published scRNAseq datasets of human atherosclerosis (*Cardiovasc Res*, 119:1676-1689). Similar to the findings in mouse scRNAseq, we observed higher *ZEB1* expression in human inflammatory macrophages, which also include IFN⁺ macrophages (Supplementary Figure S2C).
- The Reviewer highlights the importance of TREM1+ macrophages in foam cell formation, and it is noteworthy that non-conditional *Trem1* (-/-) ApoE^{KO} mice exhibit a phenotype similar to that of *Zeb1*^{ΔM}/ApoE^{KO} mice (*Nat Commun*, 7: 13151). However, it should be noted that while *ZEB1* is expressed in human foam cells, its expression levels were lower than in inflammatory ones (Supplementary Figure S2C).

D) REPLY TO REVIEWER #3

The manuscript from MC Martinez-Campanario et al., investigates the role of the transcription factor ZEB1 in macrophages in atherosclerotic plaque formation. To examine this, the authors have crossed LysM-Cre mice with ZEB1^{fl/fl} animals and subsequently backcrossed these and CRE- controls onto the APOE-KO background. They demonstrate that loss of Zeb1 from LysM expressing cells leads to increased plaque formation and suggest that this is due to the effect of ZEB1 loss in aortic macrophages. Further experiments suggest that this is regulated by impaired lipid processing in the ZEB1-deficient cells. Overexpression of ZEB1 through nanoparticle targeting reversed this effect, while ZEB1 expression levels also correlate with cardiovascular outcomes in patients. Together this is an interesting study, describing a previously unappreciated role for ZEB1, for which I congratulate the authors. However, in my opinion some conclusions are not yet fully supported by the data provided.

We are grateful to Reviewer #3 for her/his positive comments regarding the interest and novelty of the study.

1) LysM-Cre transgenic mice are used to target myeloid cells, however throughout the text the authors claim these effects are mediated by macrophages. Can the authors confirm there is no role for the other myeloid cells targeted with this CRE line e.g. neutrophils, monocytes, some DCs? If not, then I would strongly suggest the authors tone down their

claims that this is mediated by macrophages. Rather I would change the text to say myeloid cells.

We thank Reviewer #3 for the chance to clarify this important point.

- We fully concur with the comments of the Reviewer. Although the *LysM^{Cre}* mouse model has been widely used to study the role of macrophages in the atherosclerotic plaque (e.g., *Cell Metab*, 31:518-533; *Nat Commun* 14:4101; *Nat Commun* 14:4622), as the Reviewer points out, the *LysM* gene is expressed in other myeloid cells beyond macrophages. In that line, in the manuscript (page 6 of the original manuscript, page 5 in the revised manuscript), we noted that *LysM^{Cre}* targets all myeloid cells.
- We would like to note that many of the *ex vivo* and *in vitro* experiments in the study use macrophages, not simply myeloid cells. For instance, the RNAseq shown in the study (former Figure 3A, now Figure 4A) was conducted using RNA from macrophages infiltrating the atherosclerotic plaques of *Zeb1^{WT}/ApoE^{KO}* and *Zeb1^{ΔM}/ApoE^{KO}* mice with high-fat diet. Additionally, the *ex vivo* and *in vitro* experiments in Figures 2, 3, 4, and 5 to characterize phenotypically or functionally the effect of *Zeb1* ablation, we used macrophages isolated from *Zeb1^{WT}/ApoE^{KO}*, *Zeb1^{ΔM}/ApoE^{KO}*, *Zeb1^{WT}* and *Zeb1^{ΔM}* mice.
- In any case, throughout the revised manuscript, we have renamed the *Zeb1^{ΔMac}/ApoE^{KO}* mouse as *Zeb1^{ΔM}/ApoE^{KO}*,
- In addition, we replaced “macrophages” with “myeloid” in the title, and made it clear that in our *in vivo* experiments, the *Zeb1^{ΔM}/ApoE^{KO}* mouse lacks *Zeb1* expression in myeloid cells, not simply in macrophages (pages 4,6,7,15,20).

➔ EXPERIMENTS CONDUCTED: Following the comments of Reviewer #3, the revised manuscript has implemented the following changes: a) although it was already noted in the originally manuscript, the revised manuscript emphasized that in the *in vivo* experiments, *Zeb1^{ΔM}/ApoE^{KO}* have *Zeb1* deleted in all myeloid cells (pages 5); and b) we have renamed the *Zeb1^{ΔMac}/ApoE^{KO}* mouse as *Zeb1^{ΔM}/ApoE^{KO}*, replaced “macrophages” with “myeloid” in the title, and made it clear that in our *in vivo* experiments, the *LysM^{Cre}* targets expression in myeloid cells, not simply in macrophages (pages 4,6,7,15,20).

2) For the lipid accumulation & processing studies (Fig 2) the authors use peritoneal macrophages to make their claims. However, exactly how relevant these populations are to the phenotype observed in the plaques is unclear. Can the same findings be replicated in aortic macrophages?

We are grateful to Reviewer #3 for the chance to discuss this important issue.

- The very low number of intra-plaque macrophages imposes significant constraints and, in some cases, renders certain phenotypic and functional experiments unfeasible.

Consequently, to overcome this limitation, we resorted to using peritoneal macrophages for many of the in vitro experiments in the study.

- In that line, peritoneal macrophages are widely utilized as a valuable model system to investigate macrophage immunometabolism and signaling in the context of atherosclerosis. Just to cite a few articles on atherosclerosis that use peritoneal macrophages that have been published in Nature Communications in 2023: *Nat Commun* 14:929; *Nat Commun* 14:4101; *Nat Commun* 14:4622)
- Importantly, *Zeb1*^{ΔM} peritoneal macrophages exhibited a similar phenotype to aortic *Zeb1*^{ΔM} macrophages regarding lipid accumulation, as observed in TEM analyses shown in Figure 4C (aortic macrophages) compared to Figure 4I (peritoneal macrophages). Likewise, the gene expression profiles of *Zeb1*^{ΔM} peritoneal macrophages (Figure 4B) were found to be comparable to those of aortic *Zeb1*^{ΔM} macrophages (Figure 4A). These findings suggest that *Zeb1* ablation in myeloid cells has consistent effects on macrophage phenotype and gene expression both in the aortic plaque and peritoneal cavity.
- Nevertheless, a number of experiments in the study used aortic macrophages. As noted in point 1), the RNAseq shown in the study (former Figure 3A, now Figure 4A) was conducted using RNA from macrophages infiltrating the atherosclerotic plaques of *Zeb1*^{WT}/*ApoE*^{KO} and *Zeb1*^{ΔM}/*ApoE*^{KO} mice. In addition, in the revised manuscript, we examined the shares of total macrophages and CD86⁺ and CD9⁺ macrophages within the atherosclerotic plaque (see below in our reply to point 3, as well as in Figures 2B-2H).

→ Following the comments of Reviewer #3, the revised manuscript has analyzed the shares of total macrophages and CD86⁺ and CD9⁺ macrophages within the atherosclerotic plaque (Figures 2B-2H).

3) In figure 3A, the authors perform RNA-seq (bulk) analysis on aortic macrophages. However, as this is bulk analysis, this raises the question of what is being compared. Does loss of ZEB1 affect the different populations of macrophages present in the aorta? Can the authors be sure they are comparing the same cells in WT and KO? Or does the KO affect recruitment of monocyte-derived macrophages or perhaps regulate the presence of resident macrophages. Given the proposed different contributions of the various populations to the development of plaques, any heterogeneity would be important to clarify.

We thank this Reviewer for the possibility to discuss this issue. ▬

- We examined ZEB1 expression in published scRNAseq from mouse and human atherosclerosis plaques (*Cardiovasc Res*, 119:1676-1689). ZEB1 expression patterns vary among various macrophage subpopulations within both mouse and human atherosclerotic plaques. Specifically, *Zeb1* is expressed at higher levels in inflammatory

macrophages compared to other subpopulations (Figure 2A and Supplementary Figure S2C).

- As noted above, we have now employed FACS analyses to ascertain the proportions of total macrophages (CD45⁺, CD11b⁺, F4/80⁺) as well as CD86⁺ and CD9⁺ macrophages, within the atherosclerotic plaques of *Zeb1*^{WT}/*Apoe*^{KO} and *Zeb1*^{ΔM}/*Apoe*^{KO} mice (Figures 2B-2H). Notably, we observed that macrophages constituted a larger proportion of the CD45⁺ immune cell population within the plaques of *Zeb1*^{ΔM}/*Apoe*^{KO} mice compared to *Zeb1*^{WT}/*Apoe*^{KO} mice (Figure 2B). The percentage of CD86⁺ macrophages out of all macrophages was comparable in both genotypes (Figures 2C and 2E). However, consistent with the elevated macrophage infiltration in *Zeb1*^{ΔM}/*Apoe*^{KO} plaques, the proportion of CD86⁺ macrophages within the total CD45⁺ immune cell population infiltrating *Zeb1*^{ΔM}/*Apoe*^{KO} plaques exceeded that in *Zeb1*^{WT}/*Apoe*^{KO} plaques (Figures 2D and 2E).
- Regarding foam macrophages, and in line with the higher levels of foam cells in *Zeb1*^{ΔM}/*Apoe*^{KO} plaques (Figure 1I, 4C, 4I), the percentage of CD9⁺ macrophages within the overall intra-plaque macrophage population and within the entire CD45⁺ immune cell population was also higher in *Zeb1*^{ΔM}/*Apoe*^{KO} plaques compared to their *Zeb1*^{WT}/*Apoe*^{KO} counterparts (Figures 2F-2H).
- Of note, in the original manuscript, we showed that *Zeb1*^{WT}/*Apoe*^{KO} and *Zeb1*^{ΔM}/*Apoe*^{KO} mice exhibit a similar distribution of peripheral blood immune populations (original Supplementary Figure S1F, now Supplementary Figure S2A). Spleens had a larger number of immune cells than the atherosclerotic plaques, which allowed us to assess a broader range of antibodies. In our revised manuscript, we used FACS to analyze myeloid cell populations in the spleens of both genotypes and found that, within CD11b⁺ cells, the shares of CD11c⁺, F4/80⁺, Gr1⁺, and Ly6C⁺ cells are similar in both genotypes (Supplementary Figure S2B).

→ Following the comments of this Reviewer, the revised manuscript has implemented the following changes: 1) we analyzed ZEB1 expression in published scRNAseq from mouse and human atherosclerosis plaques (Figure 2A and Supplementary Figure S2C); b) examine the proportion of total macrophages and CD86⁺ and CD9⁺ macrophages within the atherosclerotic plaques of *Zeb1*^{WT}/*Apoe*^{KO} and *Zeb1*^{ΔM}/*Apoe*^{KO} mice (Figure 2B-2H) ; and c) analyzed myeloid cell populations in the spleens of both genotypes (Supplementary Figure S2B).

4) In figure 1 the authors show that these mice also have larger livers with more lipid deposits than WT controls. Is this also driven by macrophages? What happens to the different subsets of hepatic macrophages in these mice? Moreover, in Figure 5, the authors demonstrate the ZEB1 overexpression reverses the plaque phenotype but does this also reverse the systemic weight gain and the effects on the liver? How much of this phenotype can be attributed to aortic macrophages versus systemic effects? On that front,

can the authors also demonstrate that the mice consume similar quantities of food and have similar metabolic outputs as controls?

We thank Reviewer #3 for her/his comments.

We concur with this Reviewer's suggestion that *Zeb1^{ΔM}/Apoe^{KO}* mice have an altered systemic metabolic phenotype. To further define this phenotype and the mechanism of action by which ZEB1 regulates it, in the revised manuscript, we have conducted the following experiments.

- The serum levels of a panel of cytokines were examined in *Zeb1^{WT}/Apoe^{KO}* and *Zeb1^{ΔM}/Apoe^{KO}* mice using a commercial multiplexed sandwich- and bead-based quantitative antibody array commercial kit (RayPlex® Mouse Inflammation Array Kit 1, RayBiotech Life, Inc.), which was assessed by FACS. As shown in the new Figure 2M, the sensitivity of this array was able to detect 4 of the 13 cytokines included in the kit, namely IL1β, IL2, IL12, and CCL2. *Zeb1^{ΔM}/Apoe^{KO}* mice expressed higher serum levels of these cytokines compared to *Zeb1^{WT}/Apoe^{KO}* mice.
- Compared to *Zeb1^{WT}/Apoe^{KO}* livers, the livers of *Zeb1^{ΔM}/Apoe^{KO}* mice exhibited higher expression of the nuclear processed form of SREBP1c (Figure 2S). Nuclear SREBP1c triggers the activation of liver enzymes that promote *de novo* lipogenesis, leading to the accumulation of fatty acids in the liver and higher serum levels of serum lipids (reviewed in *Cell*, 184: 2537-2564, *Biochem J.* 478:3723-39).
- We also examined the potential role of ZEB1 in white adipose tissue (WAT)-mediated inflammation. WAT is a major source of systemic inflammation through the activities of adipocytes themselves as well as of infiltrating immune cells, chiefly macrophages (reviewed in *Circ Res.* 128:951-68). Compared to *Zeb1^{WT}/Apoe^{KO}* mice, the WAT of *Zeb1^{ΔM}/Apoe^{KO}* mice exhibited higher infiltration of macrophages (Figure 2L). In *Zeb1^{ΔM}/Apoe^{KO}* mice, infiltrating macrophages were effectively knock out for *Zeb1* (Figure 2J) and expressed higher *Il1b* mRNA levels compared to WAT macrophages in *Zeb1^{WT}/Apoe^{KO}* (Figure 2K).
- We found that serum levels of leptin in *Zeb1^{ΔM}/Apoe^{KO}* mice are higher than *Zeb1^{WT}/Apoe^{KO}* mice (Figure 2N). In addition to myocardium and perivascular adipose tissue, WAT (mainly its adipocytes) is a major source of the adipokine leptin (*Atherosclerosis* 189:47-60; *Nat Rev Cardiol.* 7:22-29). While leptin has pleiotropic effects in atherosclerosis in human and mice, leptin promotes the recruitment of monocytes into the arterial intima, the formation of foam cells, the proliferation of vascular smooth muscle cells, and the secretion of classical proinflammatory cytokines.

➔ Following the comments by Reviewer #2, the revised manuscript has implemented the following changes: a) we have quantified serum levels of cytokines in *Zeb1^{WT}/Apoe^{KO}* and *Zeb1^{ΔM}/Apoe^{KO}* mice (Figure 2M); b) analyzed SREBP1c in the livers of both genotypes (Figure 2S), c) examined by infiltration by macrophages of the white adipose tissue in *Zeb1^{WT}/Apoe^{KO}* and *Zeb1^{ΔM}*

/ApoE^{KO} mice as well as macrophage production of *Il1b* (Figures 2J-2L); and d) assessed serum levels of leptin in *Zeb1*^{WT}/ApoE^{KO} and *Zeb1*^{ΔM}/ApoE^{KO} mice (Figure 2N).

Minor comments

5) Fig 1C appears to be incorrectly annotated as Zeb1-macs rather than Zeb1+/-?

We thank Reviewer #3 for pointing out to the mislabeling of Figure 1C, which has been corrected in the revised manuscript.

6) Sirius red should also be quantified in Fig 1I

We thank Reviewer #3 for her/his suggestion. We have now quantified Sirius staining and included it in the former Figure 1I (now Figure 1K).

7) Line 205 (of for Bodipy) typo

We are grateful to Reviewer #3 for pointing out this typo, which has been corrected in the revised manuscript (page 11).

8) Has the effect of the CRE been determined? E.g. with CRE only controls? This is especially important for systemic effects where the effects of ZEB1 overexpression were not discussed.

We are very grateful to Reviewer #3 for this comment.

- Indeed, we carefully considered this issue before commencing the project. The extensive body of evidence from the more than 7,000 papers published using the heterozygous *LysM*^{Cre} mouse over the last 20 years indicates that this model does express *LysM* from the wild-type allele and there is no dose effect with this gene between one or two alleles. Therefore, any differences in phenotypes observed between the *Zeb1*^{WT} and *Zeb1*^{ΔM} mice or between *Zeb1*^{WT}/ApoE^{KO} and *Zeb1*^{ΔM}/ApoE^{KO} mice are not related to the *LysM* gene.
- Moreover, it is worth noting that the vast majority of articles using the *LysM*^{Cre} mouse model (e.g., X-gene flx/flx; *LysM*^{Cre} mice) used the X-gene flx/flx mouse (not the *LysM*^{Cre} mouse) as control. Furthermore, a recent review examining Cre toxicity in certain models of cardiovascular disease (*Nat Cardio Res*, 1:806-816) did not find any toxicity associated with the *LysM*^{Cre} model.
- In any case, we first contacted with the group that generated the *LysM*^{Cre} mouse in 1999, Dr. Clausen (University of Mainz, Germany) confirm our conclusions from the literature. Dr. Clausen agreed to share with you my communication with him and it is available at bclausen@uni-mainz.de

REVIEWERS' COMMENTS

Reviewer #1 (Remarks to the Author):

The authors carefully revised the manuscript with the attempt to address all of the comments. Although the authors did not show additional evidence of the carcinogenic risk of ZEB1, they listed reasonable third-party evidence that such a risk could be low. I am convinced that this work is acceptable now, provided that the authors do clearly make a risk statement to avoid misleading information to the audience.

Reviewer #2 (Remarks to the Author):

The manuscript has improved significantly, no further comments

Reviewer #3 (Remarks to the Author):

In this revised version of the manuscript, the authors have nicely addressed most of my original concerns. The remaining issues (contribution of other macrophages/systemic effects), heterogeneity of aortic macrophages have been dealt with in the text therefore the reader can now make appropriate conclusions and so I have no further concerns preventing publication of the manuscript at this time.

RESPONSE TO REFEREES

Manuscript: NCOMMS-23-03596B
First Author: MC Martinez-Campanario
Corresponding Author: Antonio Postigo

REPLY TO REVIEWER #1

The authors carefully revised the manuscript with the attempt to address all of the comments. Although the authors did not show additional evidence of the carcinogenic risk of ZEB1, they listed reasonable third-party evidence that such a risk could be low. I am convinced that this work is acceptable now, provided that the authors do clearly make a risk statement to avoid misleading information to the audience.

We greatly appreciate the positive comments from Reviewer #1.

In the revised manuscript, we added a new paragraph in the Discussion about the role of ZEB1 in cancer and a disclaimer stating that future studies are needed to ensure that prolonged treatment with nanoparticles overexpressing ZEB1 in macrophages has no deleterious effects or contributions to cancer initiation

REPLY TO REVIEWER #2

The manuscript has improved significantly, no further comments

We thank Reviewer #2 for her/his positive comments on the revised version.

REPLY TO REVIEWER #3

In this revised version of the manuscript, the authors have nicely addressed most of my original concerns. The remaining issues (contribution of other macrophages/systemic effects), heterogeneity of aortic macrophages have been dealt with in the text therefore the reader can now make appropriate conclusions and so I have no further concerns preventing publication of the manuscript at this time.

We are grateful to Reviewer #3 for her/his positive evaluation of the revised manuscript.